# Saturation of the compression of two interacting magnetized plasma toroids evidenced in the laboratory

A. Sladkov[1,15], C. Fegan [2,15], W. Yao [3,4], A. F. A. Bott[5], S. N. Chen[6], H. Ahmed [7], E. D. Filippov [8], R. Lelièvre [3,9], P. Martin [2], A. McIlvenny [2], T. Waltenspiel[3,10,11], P. Antici[11], M. Borghesi [12] ✉, S. Pikuz [13], A. Ciardi [4], E. d'Humières [10], A. Soloviev[14], M. Starodubtsev[14] & J. Fuchs [3] ✉

Interactions between magnetic fields advected by matter play a fundamental role in the Universe at a diverse range of scales. A crucial role these interactions play is in making turbulent fields highly anisotropic, leading to observed ordered fields. These in turn, are important evolutionary factors for all the systems within and around. Despite scant evidence, due to the difficulty in measuring even near-Earth events, the magnetic field compression factor in these interactions, measured at very varied scales, is limited to a few. However, compressing matter in which a magnetic field is embedded, results in compression up to several thousands. Here we show, using laboratory experiments and matching three-dimensional hybrid simulations, that there is indeed a very effective saturation of the compression when two independent parallel-oriented magnetic fields regions encounter one another due to plasma advection. We found that the observed saturation is linked to a build-up of the magnetic pressure, which decelerates and redirects the inflows at their encounter point, thereby stopping further compression. Moreover, the growth of an electric field, induced by the incoming flows and the magnetic field, acts in redirecting the inflows transversely, further hampering field compression.

To investigate magnetic field compression in a setting that is frequently encountered in the Universe, i.e., that of two interacting independent large-scale magnetic structures, we use two high-power lasers (see Methods) to generate[1,2] two independent toroidal magnetic field structures, with their fields parallel to each other and akin to short flux tubes (with field strength ~100 T [see Methods]). They are set to counter-propagate toward each other at super-Alfvénic velocity.

[1]Light Stream Labs LLC, Palo Alto, CA 94306, USA. [2]Centre for Light-Matter Interactions, School of Mathematics and Physics, Queen's University Belfast, Belfast BT7 1NN, United Kingdom. [3]LULI - CNRS, CEA, UPMC Univ Paris 06 : Sorbonne Université, Ecole Polytechnique, Institut Polytechnique de Paris, F-91128 Palaiseau cedex, France. [4]Sorbonne Université, Observatoire de Paris, Université PSL, CNRS, LERMA, F-75005 Paris, France. [5]Department of Physics, University of Oxford, Parks Road, Oxford OX1 3PU, United Kingdom. [6]ELI-NP, "Horia Hulubei" National Institute of Physics and Nuclear Engineering, Bucharest - Magurele, Romania. [7]Central Laser Facility, STFC Rutherford Appleton Laboratory, Didcot OX11 0QX, UK. [8]CLPU, 37185 Villamayor, Spain. [9]Laboratoire de micro-irradiation, de métrologie et de dosimétrie des neutrons, PSE-Santé/SDOS, IRSN, 13115 Saint-Paul-Lez-Durance, France. [10]University of Bordeaux, Centre Lasers Intenses et Applications, CNRS, CEA, UMR 5107, F-33405 Talence, France. [11]INRS-EMT, 1650 boul, Lionel-Boulet, Varennes, QC J3X 1S2, Canada. [12]Center for Plasma Physics, School of Mathematics and Physics, Queen's University Belfast, Belfast BT7 1NN, United Kingdom. [13]HB11 Energy Holdings, Freshwater, NSW 2096, Australia. [14]Independent Researcher, Nizhny Novgorod, Russia. [15]These authors contributed equally: A. Sladkov, C. Fegan. ✉ e-mail: m.borghesi@qub.ac.uk; julien.fuchs@polytechnique.edu

The top view of the experimental setup is shown in Fig. 1a. As shown by the time-resolved optical measurement (see Methods) of Fig. 1c, the plasma also expands longitudinally (along the $z$-axis) in a vacuum, forming an expanding cone of 30° half-angle around the $z$-axis[3]. The magnetic field is also advected with the plasma flow away from the target surface along the $z$-axis[4] and one can observe from Fig. 1c that the plasma expands longitudinally over more than 0.5 mm in 1.5 ns. At the outermost tip of the expansion along the $z$-axis, the magnetic field strength is lowered to ~20–50 T[4,5]. Note that, due to the fact that the expanding plasmas do not have perfectly toroidal magnetic fields, and due to the fact that the two encountering plasmas are not perfectly symmetric, the encounter does not take place in a perfect 0° shear situation. Nonetheless, the fact, as detailed below, that we observe a similar compression of the magnetic fields when comparing the experimental results and the idealized simulations shows that the departure from the ideal 0° shear situation we aim at investigating here is not significant.

In our double target configuration, the targets are distanced from each other by $\alpha$ along their normal (see Fig. 1a), and when we let the two plasma plumes interact, as shown in Fig. 1d, the optical measurement reveals that there is a clear density pile-up in between the two plasmas, and that this pile-up follows an axis that is rotated by 45° compared the target normal (see the yellow dashed line).

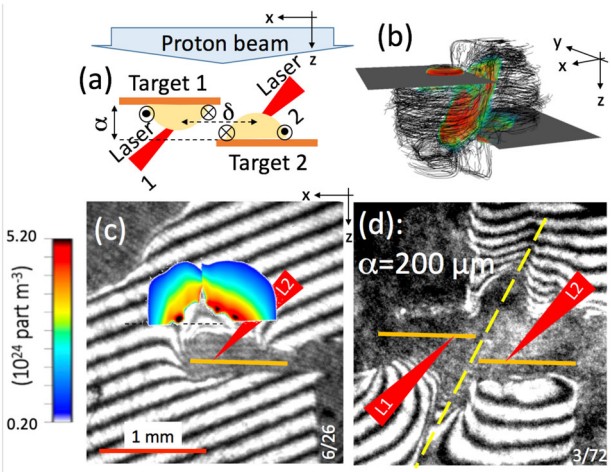

**Fig. 1 | Configuration of the plasmas used in the investigation. a** Schematic diagram of the experiment in the $xz$-plane, using two lasers (L1 and L2, separated at focus by $\delta = 500\,\mu m$ along the $x$-axis) and two targets (T1 and T2, separated along the $z$-axis by a variable distance $\alpha$). Also shown are the protons (in light blue, sent along the $z$-axis) used for the radiography diagnostic (see Methods) and the plasma plumes with frozen-in toroidal magnetic fields generated by each laser ablation at the target front (in yellow). **b** 3D simulated (see Methods) depiction of the magnetic field lines in the plasma plumes (in black), together with the compressed magnetic sheet in between the two plasmas (in color), snapshot at $t\Omega_0 = 37$. **c, d** Raw optical interferometry images of **c** just one plasma expanding from T2, and **d** the two plasmas, in the xz-plane. The dark zones correspond to the shadows induced by the initial targets and target holders, as well as the dense parts of the expanding plasmas, following the laser interaction. The fringe pattern originates from the Mach−Zehnder setup[58] we used. The deformations of the fringes encode the line-integrated plasma density accumulated as the probing laser beam propagates along the $z$-axis. As the gradient of the plasma increases toward the solid targets, the refraction of the probing laser beam increases until it cannot be collected anymore by the imaging optics, inducing the observed dark zones. The times at which the snapshots are taken are, respectively, **c** 1.5 and **d** 3 ns after the start of the laser irradiation of the targets. Panel (**c**) shows the deconvolved volumetric density map (encoded in color, see the colormap on the left), illustrating, as expected, the plasma expansion along the target normal, i.e., along the $z$-axis. In **d**, the initial locations of the T1 and T2 targets are indicated by the orange lines. Source data are provided as a Source Data file.

Table 1 summarizes the parameters of the plasma within the magnetic compressed sheet located in between the two counter-streaming plasmas of the present laboratory experiment (right column). We note that, in the experiment, the flows simply do not exist for long enough for turbulence to develop by the time at which the compression of the field is observed. A very simple estimate of the characteristic time $t_{turb}$ required for the development of turbulence can be made based on the inflow velocity and the characteristic size of the compressed region: $t_{turb} \sim L/V_{inflow} = 6$ ns, where $V_{inflow}$ is the inflow velocity and $L$ is the characteristic spatial scale (see Table 1). This timescale, which is essentially the timescale of nonlinear interactions between fluid motions in the compressed layer, is a factor of 5–6 times longer than the timescale on which peak compression is observed (~1–2 ns), and a factor of nearly three times longer than the time of the latest measurement. Furthermore, the good agreement between the experiment and the simulations, detailed below, the latter of which do not manifest turbulent flows, supports the claim that turbulence does not seem to have a major impact on compression.

A similar set of parameters that could be retrieved from satellite observations of the interaction between solar coronal mass ejecta[6], and to which our experiments can be compared, is also given in Table 1 (middle column). In order to evaluate if, indeed, such two widely different systems can be scalable to each other, we follow the approach presented in ref. 7,8. To do that, we will first evaluate if, for both systems, the convection dominates diffusion, in which case we can assert that these systems can be described in the ideal MHD framework. This is quantified by looking at the Reynolds number (the ratio of the convection over ohmic dissipation, $R_e = LV/\nu$, in which $\nu = 2 \times 10^8 T_e/(ZB)$ where Z is the ionization degree of the plasma ions), the magnetic Reynolds number (the ratio of the convection over the magnetic diffusion, $R_m = LV/\eta$, in which $\eta = 8.2 \times 10^5 \Lambda Z T_e^{-3/2}$, $\Lambda$ is the Coulomb logarithm), as well as the Peclet number (the ratio of magnetic convection over magnetic diffusion, $P_e = LV/\chi$, in which $\chi = 8.6 \times 10^9 A^{1/2} T_e/(ZB)$ where A is the

**Table 1 | Plasma parameters, within the compressed magnetic sheet located in between the two magnetized toroidal plasmas, of the laboratory experiment (present case, right column)**

|  | Coronal mass ejecta | Present case |
| --- | --- | --- |
| Inflow velocity $V_{inflow}$ [km/s] | 250 | 50 |
| B (T) | 3e-8 | 450 |
| Electron density [cm$^{-3}$] | 2 | 1.3e21 |
| Sound mach number ($M_s$) | 1.4 | 3 |
| Alfven-mach number ($M_A$) | 1.6 | 10 |
| Characteristic spatial scale ($L$ [cm]) | 1.5e12 | 3e-2 |
| Reynolds number ($R_e$) | 5e5 | 1.2e3 |
| Magnetic Reynolds number ($R_m$) | 1.5e17 | 63 |
| Peclet number ($P_e$) | 1e4 | 3.5 |
| Measured magnetic compression ratio | 2.8 | 8 |
| Expected magnetic compression ratio | 3.2 | 7 |
| Thermal beta | 0.18 | 0.1 |
| Euler number | 1.8 | 3.7 |

An extended version of the Table with more parameters is given in the Supp. Info. In the middle column are given the plasma parameters that can be retrieved from satellite observations of an interaction between coronal mass ejecta (CME)[6]. The plasma parameters, in this case are per-taining to the compressed magnetic sheet resulting from a later CME catching up earlier and slower CMEs. Note that $M_A$ in both cases is calculated using the magnetic field strength of the inflow (i.e., in the foot region, which is 10 nT for the CME and 60 T for the laboratory experiment). The expected magnetic compression ratio is calculated on the base of the maximum value of the compressed magnetic field being $B_{max} = V_{inflow}\sqrt{\mu_0 m_i n_i}$ where $\mu_0$ is the vacuum permeability and $n_i$ (resp. $m_i$) is the ion density (resp. mass), see text for details.

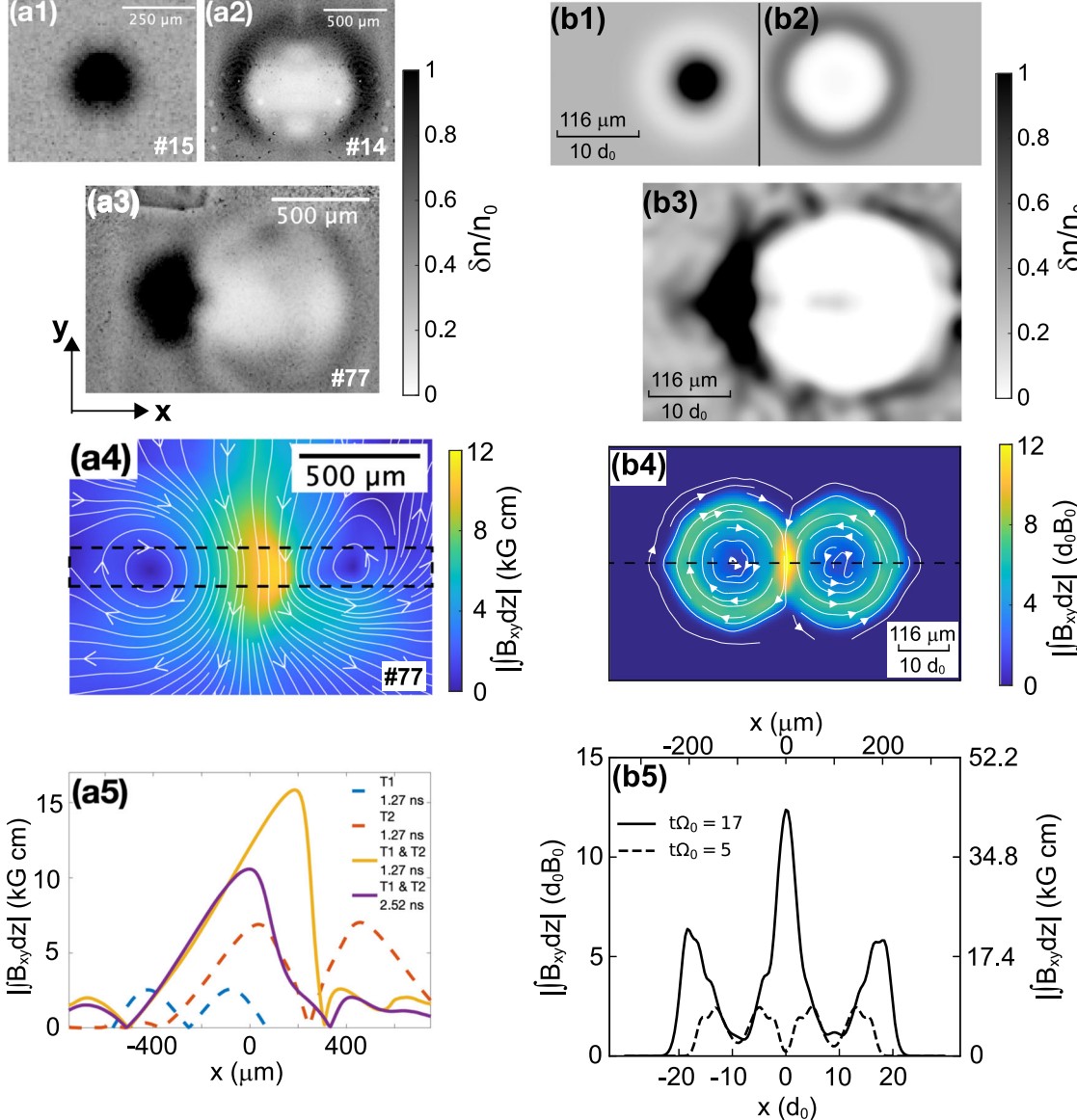

**Fig. 2 | Experimental and simulated evidence for magnetic compression. a1–a2** Experimental proton-radiography images (see Methods) probing the single toroidal magnetic field in the plasma plume produced on **a1** target T2 or **a2** target T1. Both images are snapshots taken at 1.27 ns. **a3** Same but when there are the two targets (snapshot at 2.52 ns, with $\alpha = 200$ μm). **a4** Path-integrated magnetic field strength analyzed via the code PROBLEM[13]. The white-arrow streamlines represent the in-plane magnetic field lines ($B_x$ and $B_y$), and the colormap shows the path-integrated (along the $z$-axis) strength of the $xy$-plane magnetic field. **a5** Lineout, along the $z$-axis, of the path-integrated magnetic field strength (measured in the black dashed box shown in panel (**a4**) and averaged along the $y$-axis). Also shown are the lineouts corresponding to the images shown in panels (**a1**–**a2**) and for

another shot (#29) where the two plasmas interact at an earlier time, i.e., 1.27 ns. The right column shows the corresponding results obtained from the hybrid simulations (see Methods for details and for the normalization factors). Correspondingly to the experimental images, the $\alpha$ parameter is set to $20d_0 = 232$ μm, and the snapshot is taken at $t = 17\Omega_0^{-1} = 2$ ns. The synthetic proton dose distributions are shown for the two targets case (**b1**–**b2**) before and (**b3**) after their interaction. **b4** The path-integrated magnetic field strength distribution in the $xy$-plane of the simulation box, along with (**b5**) the lineout taken along the black dashed line (at $y = 0$) shown in panel (**b4**). The lineout taken at a very early time ($t = 5\Omega_0^{-1} \sim 0.6$ ns), before the two individual toroids interact, is also shown. Source data are provided as a Source Data file. Source data are provided as a Source Data file.

mass number of the plasma ions). As shown in Table 1, we quantitatively verify these two parameters indeed are, for both systems, larger than one. Further, as proposed by ref. 9, we can calculate the two scaling quantities, the Euler number ($E_u = V(\rho/p)^{1/2}$) and the thermal plasma beta ($\beta = 8\pi p/B^2$), where $\rho$ is the mass density, and $p$ is the plasma pressure $p = k_B(n_i T_i + n_e T_e)$. In the case these two quantities are found to be similar for the two systems, we can then indeed assert that the two systems can be scaled to each other and evolve in the same manner. We can observe from Table 1 that, indeed, for both systems these numbers are quite close. From this, we can deduce that the two systems are indeed scalable from one to the other.

To characterize the individual magnetic field structures produced by each plasma in the laboratory experiment, as well as the compression produced by their encounter in the region between the two targets, we use proton radiography[10] (see Methods). This diagnostic yields the magnetic field distribution in strength and spatial distribution in the $xy$-plane.

## Results

The experimental films shown in Fig. 2a1–a3 display the dose modulations recorded by 6.6 MeV protons of the magnetic fields on a single T1 target (a1), on a single T2 target (a2), and when both T1 and T2

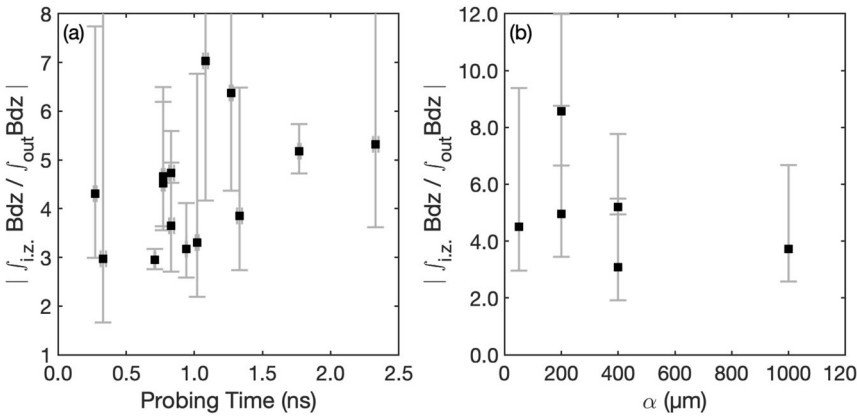

**Fig. 3 | Variation of the experimentally measured magnetic compression as a function of time and space. a** Temporal evolution of the ratio of the path-integrated (along the $z$-axis) magnetic field strength in the interaction zone (i.z., corresponding to the maximum recorded in maps similar to the one shown in Fig. 1 (**a4**)) to the path-integrated magnetic field strength in the non-interacting outer region (out). For the latter, we take the average between the values recorded on the left and right of the interaction zone (corresponding to the outer parts of the involved toroidal magnetic fields). The error bars correspond to taking the extrema between the outer magnetic fields recorded on the left and on the right. The data shown here correspond to a target separation $\alpha = 100\,\mu m$ and are formed by amalgamating different proton radiographs across multiple laser shots. **b** Evolution of the same ratio when increasing the target separation, $\alpha$. These values were extracted from different shots, all corresponding to probing the plasmas with 4.43 MeV energy protons, at 2.58 ns after the peak of the long pulse lasers arrived on target. Note that, but only for the shots used for panel (**b**), we reduced the on-target intensity of the L1 and L2 lasers (to ~$10^{14}$ W/cm²), thus showing also the robustness of the results with respect to such change in the initial conditions. Source data are provided as a Source Data file.

targets are present (a3). Note that in the latter case, we have chosen the separation between the targets $\alpha$ (here equal to 200 μm) to be such that it is smaller than the longitudinal (along the $z$-axis) extent of a single-plasma plume for the times considered here, as inferred from the optical probing data. This allows us to make sure of the overlap between the two plasma plumes. To demonstrate the robustness of the observed magnetic field compression, a complementary dataset obtained for $\alpha = 400\,\mu m$ is shown in Fig. S1 of the Supp. Info. Since the direction of the magnetic fields from the two targets relative to the probing proton beam are opposite (see Fig. 1a), therefore the magnetic fields affect the probing proton beam in opposite ways. When there is only a plasma expanding from the T2 target, the magnetic field structure focuses the proton beam, leading to a concentrated proton dose (see Fig. 2a1), as expected[11,12]. Conversely[11,12], the proton deflection pattern is reversed when there is only a plasma expanding from the T1 target, as the magnetic field structure now defocuses the proton beam, yielding a radiograph characterized by a large white ring structure surrounded by a dark ring, the probing protons having been expelled from within, and to be accumulated at the edge (see Fig. 2a2). Now, when the two plasmas are simultaneously expanding from the T1 and T2 targets, the proton deflection pattern differs quite significantly from what would be the simple linear overlap of two single-plasma-induced patterns. Indeed, as can be seen in Fig. 2a3, the resulting pattern is such that the focused proton structure has now largely expanded into an arc shape, whereas the defocused white ring is vertically stretched with a clear disruption at the top and bottom.

Such a pattern could suggest that there is a compressed region, namely an increased strength magnetic field in the two plasmas interaction region. If we assume that the probing protons propagate through the compressed magnetic field region, we can expect that they are deflected more toward the focused spot. Compared to the single-beam case shown in Fig. 2a1, a2, the proton deflection pattern we observe in Fig. 2a3 is no longer azimuthally symmetric because compression occurs only in a small region. For the same reason, the defocused ring structure is broken, with the curvature radius of the resulting structure being larger than that produced by the single-plasma plume.

To test this hypothesis, Fig. 2a4 displays the path-integrated magnetic field map reconstructed from the proton deflectometry map shown in panel (a3) using the PROBLEM algorithm[13] (see Methods).

Here, we can clearly distinguish the two individual plasma plumes with opposite polarity, and the magnetic field build-up in between. The magnetic fields in the two plasma plumes are of similar shape and strength, as expected, since this is what we observe for the non-interacting single-plasma plumes corresponding to panels (a1, a2). Figure 2a5 shows lineouts of the path-integrated magnetic field from the map of the interacting plasma plumes (panel a4, corresponding to a probing time of 2.52 ns after the start of the laser irradiation), as well as that from another shot taken at an earlier time (1.52 ns). The same panel also shows lineouts from the individual plasma plumes reconstructed from panels (a1) and (a2). Note that the data for the individual plasma plumes are taken early on (1.27 ns), in order to show that the magnetic field is then already fully developed and present at the interaction point ($x = 0$). Despite the shot-to-shot variation, one can quantitatively discern the compression of the magnetic field in the interaction zone, compared to the field in the individual plasma plumes on both sides of the plasma plumes encounter ($B_{foot}$, in reference to the standard terminology used for magnetic field compression in shocks[14]).

To quantify the level of maximum compressed magnetic field ($B_{max}$), we have to consider that the effective compression in the three-dimensional plasma plumes takes place only over a fraction ($h$) of the maximum possible interacting length $\alpha$ (see Fig. 1a). Thus, one can write in the interaction zone (i.z.): $\int_{i.z.} Bdz = B_{max}h + B_{foot}(\alpha - h)$. Since we have, for the non-interacting outer region, $\int_{out} Bdz \approx B_{foot}\alpha$, it leads to:

$$B_{max}/B_{foot} \approx 1 + \left( \int_{i.z.} Bdz \middle/ \int_{out} Bdz - 1 \right)(\alpha/h).$$

We infer from Fig. 2a5 that ($\int_{i.z.}Bdz/\int_{out}Bdz$) ~2–4, while our simulations detailed below suggest that $\alpha/h$ ~2–4, thus yielding a compression level of $B_{max}/B_{foot}$ ~ 3–13. We stress that the data shown in Fig. 2 is only a subset of the overall data taken during our experiments, in all of which we have observed the same range for the maximum magnetic compression. For example, Fig. 3 shows variations of the raw ratio of the magnetic field strength measured at the center, in between the two interacting toroidal magnetic fields, over the incoming magnetic field (as measured in the outer part of the toroids). The variations correspond to probing the plasmas, and the interactions between the two

toroids, at various times after the start of the laser irradiation of the targets (see Fig. 3a) and to probing the interaction for various separations between the two targets (see Fig. 3b). Note that the values of the raw ratio $\int_{i.z.} B dz / \int_{out} B dz$ is between ~3 and ~9, which translates into an average value for $B_{max}/B_{foot}$ ranging between ~7 and 25.

## Discussion

The limited magnetic compression observed in the experiments could seem surprising, given that much higher compression ratios, i.e., up to thousands, could be obtained when an overall distributed magnetic field is compressed, e.g., radially or by a shock[15–17]. To investigate the dynamics underlying the observed limited compression, we performed numerical simulations with the three-dimensional (3D) hybrid particle-in-cell (PIC) code AKA[18–20] (see Methods). In order to have the simulation computationally manageable, we decreased both the initial radius of each toroid and the distance between the toroids by a factor of ~2 with respect to the experiment. We believe that such scaling has, however, weak consequences on the physics: having larger magnetic loops would decrease the curvature radius of the magnetic field lines at the compression site, but it will not change the micro-physics at play in the compression process. Just as in the experiments, we can simulate one single-plasma plume, or the interaction between two plasma plumes. The simulation program also has the capability of producing synthetic proton radiographs. These, as shown in Fig. 2b1–b3, are in good agreement with the features observed in the experiments, as are the magnetic field map and lineouts shown in Fig. 2b4–b5. We note that we have a factor 2 difference in the absolute value of the compressed magnetic field between the simulations and the experiment, but as detailed below, the compression ratio itself, i.e., $B_{max}/B_{foot}$ is well matched between the experiment and the simulation.

As we will now detail, the limited compression can be mostly understood in the frame of an ideal MHD. Fundamentally, that limitation is the consequence of an induced electric field ($\mathbf{V}_{inflow} \times \mathbf{B}$, in which $\mathbf{V}_{inflow}$ is the flow velocity and $\mathbf{B}$ is the magnetic field) that is present on both sides of the compressed magnetic sheet. This field arises following the penetration of each plasma plume in the magnetic field of the opposite plume. This then induces a $\mathbf{V}_{inflow} \times \mathbf{B}$ electric field, which is directed along the $s$-axis (i.e., parallel and antiparallel to it), see Fig. 4a. This field affects the plasma transport, redirecting the flows, in a similar manner as documented in space plasmas by satellite observation near Earth[21]. Then, the plasmas flowing along the $s$-axis also induce an electric field, illustrated by the black arrows in Fig. 4b. The resulting plasma drift $\mathbf{E} \times \mathbf{B}$ is directed up/down along the slanted compressed magnetic sheet (i.e., along the $s$-axis). Since the induced electric field acts to deflect the inflows coming onto that sheet, it, therefore, limits further compression of the frozen-in magnetic field. It is also this deflection that induces the slanted pile-up density structure that is experimentally observed in Fig. 1d.

The same slanted structure is observed in the simulations, as shown in Fig. 4a that displays the $y - z$ map of the interacting plasma plumes at the end of the simulation, i.e., at $t = 50 \Omega_0^{-1} = 5.88$ ns for $\alpha = 20 d_0 = 232$ μm. There, we can discern the two sources of the magnetic toroids (located at the surface of the two targets), their expansion driven by that of the plasma, and the compressed magnetic sheet at the interface between the two plasma plumes.

Figure 4c-left panel shows that the electrons are highly magnetized near the magnetic sheet: there, the plasma beta parameter for the electrons (i.e., the ratio of the electron pressure to the magnetic field pressure) stays <1. This is not the case for the ions (see the right panel of Fig. 4c), as they transform their ram pressure into thermal pressure. This is shown in Fig. 4d, which displays the ratio of the ion ram pressure $\rho V_{inflow}^2$, where $\rho$ is the plasma density, to the magnetic pressure. The underlying mechanism is the deceleration of the ions by the electric field arising because of the gradient of the magnetic pressure. The ion magnetization is quantified in Fig. 4e (see the black dashed line). It

shows that the ion Larmor radius is decreasing, and becomes, at the center of the compressed magnetic sheet, $R_L(r = 0) \approx d_i$, where $d_i$ is the local ion inertial length.

Once the ions are magnetized and cannot leave the sheet, they are subjected to the induced electric field, $-\mathbf{V}_{inflow} \times \mathbf{B}$, and merely drift in the outflow direction (the $s - axis$ in Fig. 4a). This is illustrated in Fig. 4b, which shows the (integrated along the $y - axis$) electron density, together with the electric fields (black arrows), and ion flow velocities (black arrows). The ion dynamics, as they approach the magnetic sheet, is illustrated in Fig. 5, which represents the ions in the phase space $(r, V_r)$. In the single-plasma plume case (Fig. 5a), i.e., when there is no magnetic sheet hindering the ions flow, we observe that the $V_r$ velocity of the ions stay quite constant. On the contrary, in the case of the interacting plasma plumes (Fig. 5b), we observe that the ions flow $V_r$ velocity reduces on both sides of the magnetic sheet, as the ions are effectively redirected along the sheet (see the black arrows in Fig. 4b) by the associated electric field. Additionally, comparing the ions distribution functions for the cases with (Fig. 5c) and without (Fig. 5b) the Hall term included in the modeling of Ohm's law (see Methods), we notice in the latter case the increase of the deceleration and of the reflection of the ions inside the sheet. Although the ideal MHD framework can capture the essence of the dynamics, non-ideal effects render the magnetic field accumulation and its effects on the ion flow stronger.

To estimate the compressed magnetic field strength $B_{max}$, we first observe that, as shown by the full black and dotted black lines in Fig. 4e, the quasi-stationary compressed magnetic sheet can be very well approximated by the stationary solitary solution[22] ~$0.2 + B_{fit} cosh^{-2}(r/\lambda)$, i.e., a compressed magnetic field peaked in the center (at $r = 0$), on top of a foot at $0.2 B_0$. From this, and as expected[14,23–25], we can measure in Fig. 4e that the width (at half-maximum) of the magnetic sheet is close to the ion inertial length. Moreover, based on the measurement that $R_L(r = 0) \approx d_i$, and since we have $R_L = V_{inflow}/\Omega_{ci}$ where $\Omega_{ci}$ is the ion cyclotron frequency and $d_i = V_A/\Omega_{ci}$, where $V_A$ is the (local) Alfvén velocity, we can thus write that the Alfvén velocity becomes approximately equal to the plasma flow velocity at the compressed edge of the magnetic sheet. From this, we deduce that $B_{max} = V_{inflow}\sqrt{\mu_0 m_i n_i}$. To quantitatively evaluate $B_{max}$, we evaluate the point in the flow where the ions start to be magnetized, the flow velocity reduces, and the magnetic field starts to grow. We evaluate such an edge of the compressed magnetic sheet to be located around $r = 0.85 d_0$, which corresponds to half of the sheet thickness. There, $V_{inflow} \approx 0.4$ (not shown). Since we also have $n_e/n_0$ ~4 and thus $n_i/n_0 = n_e/(Z n_0)$ ~0.2, with Z = 18 being the ionization of Cu (of $m_i = 64 m_p$) at that location, we finally obtain $B_{max}/B_0$ ~ 1.4. Since $B_{foot}/B_0 = 0.2$, it then yields $B_{max}/B_{foot}$ ~7, i.e., not only consistent with what is directly observed in the simulation (see Fig. 4f), but also very close to the experimentally evaluated compression (~8). Alternatively, we can derive the value of the compressed magnetic field by considering that the energy is conserved in the system. Initially, the energy is partitioned between that of the unperturbed magnetic field with pressure $0.5 B_{foot}^2$ and that of the plasma with ram pressure $\rho V_{inflow}^2$. In the compressed magnetic sheet, all the energy is mostly contained within the magnetic field, with pressure $0.5(B_{max})^2$. Thus, the energy conservation yields $B_{max}/B_{foot} = \sqrt{1 + 2 M_A^2}$ where $M_A = V_{inflow}/V_A$ and $V_A = B_{foot}/\sqrt{\mu_0 m_i n_i}$. Since we have $M_A$ ~10 for the inflow (see Fig. 4e), we can deduce a maximum magnetic compression ratio of $B_{max}/B_{foot}$ ~14, i.e., of the same order as the previous estimate. Note that since the Mach number is ≫ 2, i.e., that we are in a supercritical regime[14,24], we indeed expect that a substantial part of the ions are turned around by the magnetic piston. All this shows that the limited compression we observe is robust, since we have a good agreement between the experiment and the simulation. We have pinpointed a mechanism that

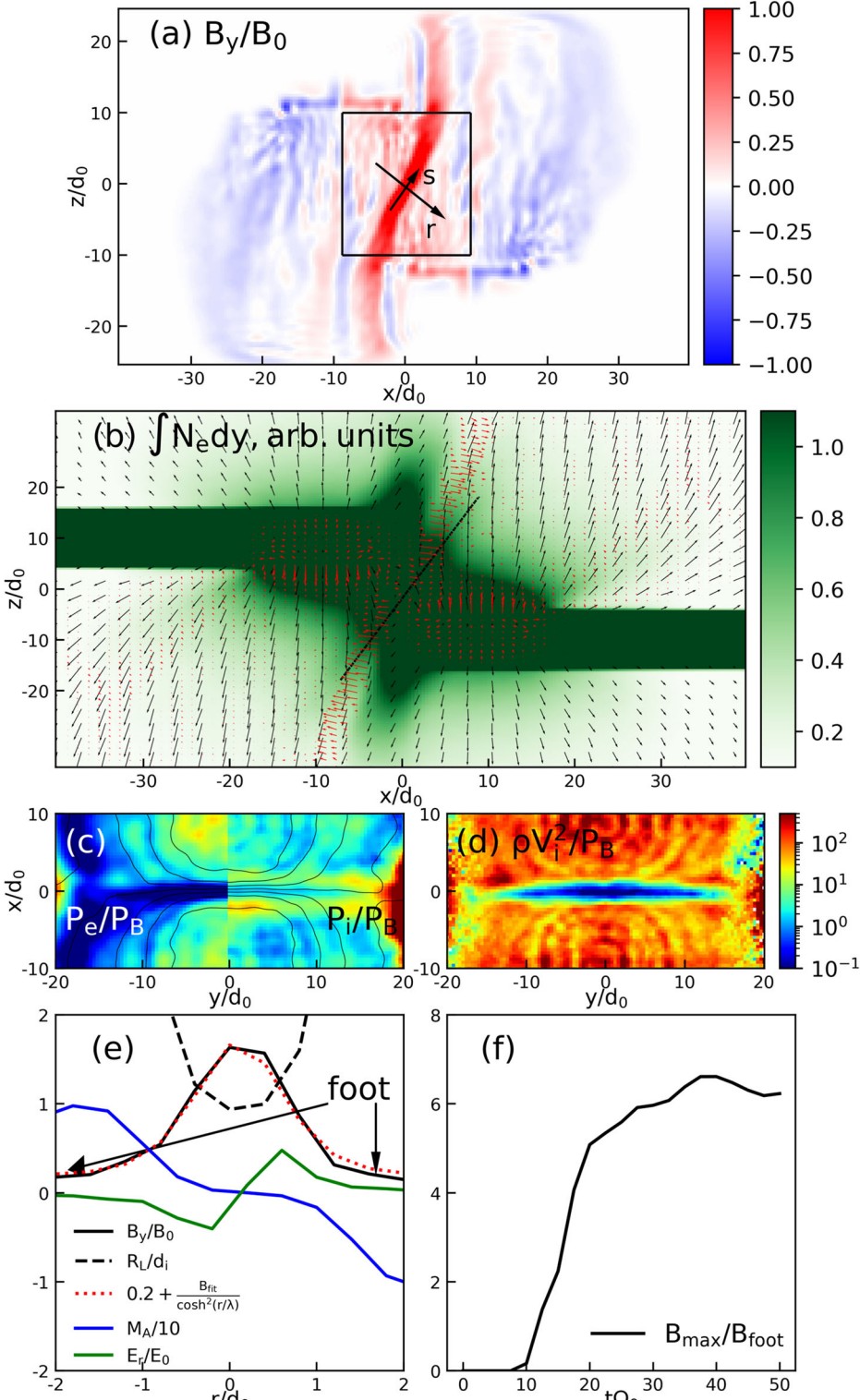

**Fig. 4 | 3D simulations and analysis of the magnetic compression.** Simulation results of (**a**) $B_y$ (normalized to $B_0$, see Methods) in the two interacting plasma plumes with their magnetic fields at t = $50\Omega_0^{-1}$ = 5.88 ns for $\alpha = 20d_0 = 232$ μm, **b** integrated electron density (normalized to $n_0d_0$, see Methods) at the same time as in panel (**a**), but for $\alpha = 10d_0 = 116$ μm with superposed associated electric fields (black arrows) and ion flows (black arrows); the black line marks the integrated density in between the two plasmas, which is markedly slanted, similarly as in the experimental image of Fig. 1d, c thermal plasma beta (left part $y < 0$ - ratio of the thermal electron pressure to magnetic pressure; right part $y > 0$ - ratio of the

thermal ion pressure to magnetic pressure) with superposed magnetic field lines in black, and (**d**) ram plasma beta (i.e., ratio of the ion inflow ram pressure to the magnetic pressure). Both (**c**, **d**) are displayed at $t\Omega_0 = 37$ for $\alpha = 20d_0$. **e** Lineouts along the auxiliary axis $r$ (see panel (**a**)) of the Alfven-Mach number divided by 10, in blue), electric field $E_r$ (in units of $V_0B_0$, green), ratio of the ion Larmor radius $R_L$ to the local ion inertia length ($d_i$), accumulated magnetic field $B_y$ (in units of $B_0$, black) and an approximation by the solitary solution (in black dotted line) $0.2 + B_{fit}cosh^{-2}(r/\lambda)$ with $\lambda/2 = 0.85$ (in units of $d_0$). **f** Time evolution of the magnetic compression for $\alpha = 20d_0$. Source data are provided as a Source Data file.

can explain the very modest compression factor observed at very diverse scales in the Universe during the interaction of two magnetic field structures advected by matter, whether galactic or star-emitted (as exemplified by the direct comparison detailed in Table 1 between the experimental plasmas and those of interacting solar CMEs). Magnetic fields are not only a major source of energy in many processes, such as jet formation[26] or cosmic-ray acceleration[27], but also affect fundamental processes like thermal conduction and radiative cooling in optically thin plasma[28], and as well modify the dynamics of astrophysical objects at every scale, ranging from the formation of stars[29,30] and galaxies[31] to the dynamics of accretion disks[32,33] and solar phenomena[34,35]. We thus anticipate that our results will help to enhance the comprehension of all these phenomena, through an understanding of the dynamics of the magnetic field regulating them. Indeed, when two plasmas carrying magnetic field interact, e.g., in configurations as varied as galaxy-cluster[36], supernova remnant-cloud[37], or wind-exoplanet[38] configurations, the overall magnetic field will be affected, as analysed here, thereby impacting the future evolution of the system.

Since the present investigation was set in a configuration where the two magnetic structures have field lines parallel to each other, i.e., with mostly zero shear angle, a direction to investigate other configurations encountered in space and astrophysical plasmas will be to observe the transition from magnetic compression, observed here, to reconnection[39], corresponding to a 180 degrees shear angle between the field lines. This would allow one to observe the progressive weakening of the compression measured here to a growing annihilation of the magnetic field, being maximized[5] for perfectly antiparallel encountering magnetic fields. Doing this could be achieved simply by rotating the plane of one target with respect to the $x$-axis.

This laboratory also approach bears promise to extend the investigation to the strong magnetic compression suggested by observations[40] in the collisionless relativistic plasmas of gamma-ray bursts, which is thought[41,42] to be at the source of the large and highly energetic synchrotron emissions observed to originate from them. This should now be possible using existing ultra-intense lasers[43], capable of producing highly relativistic electrons[44] and ultra-strong magnetic fields[45,46].

## Methods

### General experimental setup

The experiments were performed at two different high-intensity laser facilities, namely LULI2000 (France) and VULCAN (Rutherford Appleton Laboratory, U.K.), which both have similar laser parameters, in order to field complementary diagnostics on the interested plasma interaction. We note that in a previous experiment[47], using a similar configuration, they were not able to measure any magnetic field compression, likely due to the short duration over which the magnetic fields were generated and a too large distance between the magnetic fields, thereby weakening magnetic compression. In our experiment, two 5-µm-thick copper targets (T1 and T2) were irradiated by two laser beams (L1 and L2, having ~35 GW power over up to 5 ns and focused over azimuthally averaged radii of ~25 µm circular focal spots, yielding on-target intensity of ~$10^{15}$ W/cm²). The laser irradiation on the two targets generated two hot, dense plasmas that expand radially toward each other. A time-integrated x-ray spectrometer with spatial resolution (FSSR)[48], which records L-shell lines from the copper plasma (see below), allowed us to determine the peak electron temperature in the laser-irradiated region to be ~300 eV with an average ionization level of 19. Regarding the separation between the two targets, we note that, as previously recorded[49] using similar laser conditions, the radial magnetic field expansion is ~300 µm/ns. Therefore, we chose the separation between the two laser impacts to be, along the $x$-axis, $\delta = 500$ µm (see Fig. 1a), such that there is overlap between the two magnetic fields associated with the plasma plumes and within the first

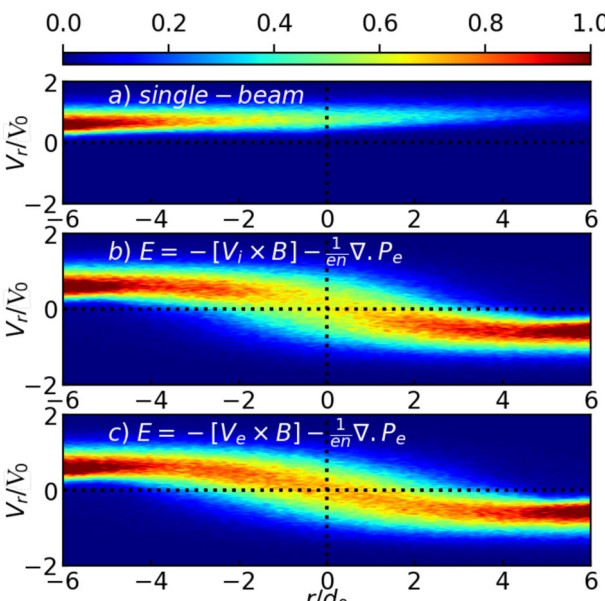

**Fig. 5 | Phase space ($r$,$V_r$) maps as the particles, as retrieved from the simulations.** The data correspond to what is recorded in the black rectangle in Fig. 4a at $t = 50\Omega_0^{-1} = 5.88$ ns. The coordinates ($r$,$s$) correspond to the two perpendicular axes in the plane $x - z$ shown in Fig. 4a, i.e., the unit vector $r$, which is along the normal to the magnetic sheet, and $s$, which is along the magnetic sheet. The particle number represented by the colorbar is normalized to $N_{max}$. Shown are: **a** the single-beam case, **b** the two targets case without the Hall effect in Ohm's law, and **c** the two targets case with the Hall effect included. Source data are provided as a Source Data file.

ns of the magnetic field generation. In each plume, the density and temperature gradients generate the so-called Biermann-Battery magnetic fields[1,2]. Detailed characterization[5,49,50] of the magnetic fields were performed in similar experimental conditions and had shown that the overall topology of each magnetic field is analog to a short flux tube connected to itself (see Fig. 1a, b). Around the laser energy deposition zone, the magnetic field is further compressed toward the target by the Nernst effect[49,50], reaching strength ~100 T for the entire laser pulse duration.

### Optical interferometry

The plasma electron density is recorded by optically probing the plasma (using a milliJoule energy, 10 ns duration, 527 nm wavelength auxiliary laser pulse), coupled to Kentech gated optical imagers, in order to have a snapshot of the plasma over a duration of 100 ps, and using a standard Mach–Zehnder interferometry setup. It allows to measure electron plasma densities in the range of $10^{17}$ to $10^{19}$ cm⁻³. Note that in the images, the dark zone located close to the targets is due to the refraction of the optical probe. Therefore this diagnostic method does not provide information about the dense plasma close to the surface of the target.

### Proton radiography

A short pulse, CPA laser capable of delivering ~50J, ~$10^{19}$ W/cm² was incident on 25 µm aluminum or Mylar targets to create a broadband, divergent proton beam through a sheath-acceleration (TNSA) mechanism[10]. This proton beam was the probe that was used to sample[10] the magnetic fields generated in the system. A radiochromic film (RCF) stack consisting of layers of Gafchromic HDV2 and EBT3 was used as the radiography detector. The distance between the proton source and midpoint between T1 and T2 was 9.66 mm. Additionally, the distance between the copper targets and the RCF stack was 90 mm, giving a geometric magnification of ~10.3. Separate RCFs of the same

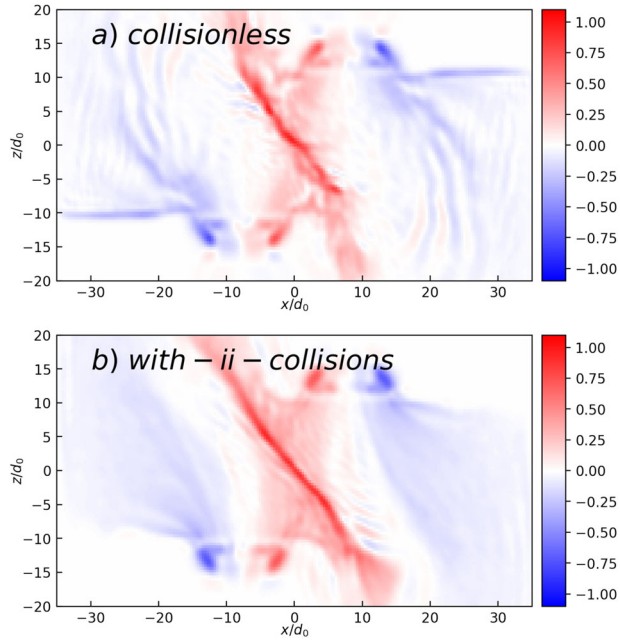

**Fig. 6 | Comparison of simulation results with and without binary ion-ion collisions.** $B_y$ component of the magnetic field at $t\Omega_0 = 50$ for the modeling (**a**) without ion-ion collisions and (**b**) with moderate ion-ion collision rate. Source data are provided as a Source Data file.

type were calibrated by physicists of the Central Laser Facility (England) using the University of Birmingham cyclotron. Through scanning the calibrated and experimental RCFs on the same scanner (EPSON Precision 2450), a model relating the optical density and dose was determined, allowing all experimental RCFs to be converted into proton-deposited dose.

## Magnetic field retrieval using PROBLEM

The proton radiography analysis code PROBLEM (PROton-imaged B-field nonlinear extraction module) was used to extract the path-integrated magnetic field from the experimental RCFs[13]. Fields are retrieved through solving the logarithmic parabolic Monge-Ampère equation for the steady-state solution of the deflection field potential. This is achieved through using an adaptive mesh and a standard centered second order finite difference scheme for the spatial discretization and a forward Euler scheme for the temporal discretization[51]. The perpendicular deflection field and, therefore, the path-integrated magnetic field can then be calculated using the solution to the Monge-Ampère equation[13]. PROBLEM can only provide a unique solution provided there are no caustics. The contrast parameter, $\mu$, was calculated using the equation found in[52] and was determined at a maximum (in the amplified region due to compression) to be $\mu$ ~0.6. Radiography was therefore conducted in either linear or nonlinear injective regimes.

## X-ray spectroscopy

A focusing spectrometer with a spatial resolution (FSSR) was used for x-ray measurements. The FSSR was equipped with a spherically bent mica crystal with a lattice spacing $2d = 19.9149$ Å; and a curvature radius of $R = 150$ mm. The crystal was aligned to operate at $m = 2$ order of reflection to record the L-shell emission spectra of multicharged copper ions in the range of 8.8–9.6 Å; (1290–1410 eV corresponding energy range). Spatial resolution $\delta x = 120$ μm along the compression axis was achieved. The spectral resolution was higher than Å/dÅ = 1000. The spectra were recorded on a fluorescence detector Fujifilm Image Plate (IP) of TR type which was situated in a cassette holder

shielded from the optical radiation. The aperture of the cassette was covered by a PET filter (2 μm thickness) coated by a thin Al (160 nm) layer to avoid the optical emission irradiating the IP. Additionally, the face of the crystal was covered by a similar filter to protect the crystal from laser-matter interaction debris and to subtract the contribution of other reflection orders to the x-ray spectra. For the measurements of the plasma parameters, we used 4d-2p and 4s-2p transitions in Ne-like ions as well as Na-like satellites which are sensitive to variations in the electron temperature and density. This emission was simulated in a steady-state approach by the PrismSPECT code[53]. A shot with two targets and large separation $\alpha$ (i.e., when the interaction between the two plasmas is negligible) was used to retrieve the information about the initial conditions for each plasma. In this case, we have the lowest contribution of the colder plasma zones at later stages of the evolution to the time-integrated spectrum, since this emission is blocked by the neighbor target. The Bremsstrahlung temperature was measured by fitting the spectral continuum with a theoretical profile based on the formula[54]: $dN/dE \sim A/\sqrt{T} \cdot exp(-E/kT)$.

## Modeling by the code AKA

We use the arbitrary kinetic algorithm (AKA) hybrid code[18], built on general and well-assessed principles of previous codes[55], such as Heckle[56]. The simulation model treats the ion kinetic dynamics following the PIC formalism and describes the electrons by a 10-moment fluid (having a density which is equal to the total ion density by quasi-neutrality, a bulk velocity, and a six-component electron pressure tensor). The magnetic field and the density are normalized to $B_0$ and $n_0$, respectively. The times are normalized to the inverse of ion gyrofrequency, $\Omega_0^{-1} \equiv m_i/(Z_i e B_0)$, in which $Z_i$ is the ionization state, $e$ is the elementary charge, and $m_i$ is the ion mass. The lengths are normalized to the reference ion inertial length $d_0 \equiv c/\omega_{pi}$, in which $c$ is the speed of light, $\omega_{pi} = (n_0 Z_i^2 e^2/m_i/\varepsilon_0)^{1/2}$ is the ion plasma frequency, and $\varepsilon_0$ is the permittivity of free space. The velocities are normalized to the Alfvén velocity $V_A \equiv B_0/(\mu_0 n_0 m_i)^{1/2}$, in which $\mu_0$ is the permeability of free space. Mass and charge are normalized to the ion ones. The normalization of the other quantities follows from these ones.

We observe in Table 1 that, with the plasma parameters used in the experiment, the dimensionless Reynolds ($R_e$), magnetic Reynolds ($R_m$), and Peclet ($P_e$) numbers are much larger than unity Thus, the plasma flows are well approximated by the ideal MHD framework. Consequently, the advective transport of momentum, magnetic field, and thermal energy are dominant over diffusive transport. To mimic the ablation process, we use a heat operator pumping electron pressure in the near-surface region of the targets, and a particle creation operator that sustains the constant solid target density which is equal to $n_0$. The ions have, as in the experiment, charge number 19 and mass number 64. These two operators create and sustain an axial electron density gradient and a radial electron temperature gradient. As a result, toroidal Biermann-battery magnetic fields are continuously produced and transported to the interaction region, where the ion flows are pressing the parallel fields against each other. The magnitude of the heat operator is adjusted to obtain the desired temperature for both ions and electrons ($T_{spot}^{e,i} = 1T_0$, which is ~ 170 eV for the reference magnetic field $B_0 = 300$ T and density $n_0 = 1.3 \times 10^{21}$ cm$^{-3}$, as inferred from the experimental measurements). With these parameters, we have $\Omega_0^{-1} = 0.117$ ns and $d_0 = 11.6$ μm. The FWHM of the heated area is $8d_0$ (~90 μm). The distance between the focal spot centers is set to $18d_0$ (~200 μm), and the $\alpha$ parameter is varied. The ion-ion collisions are taken into account with the binary collision model of ref. 57. Figure 6 shows the difference that can be observed in the simulations in the magnetic field transport for the cases without (a) and with (b) ion-ion collisions. We see that the collisions help to stabilize the magnetic sheet and allow for additional magnetic field advection along the sheet.

## Data availability

All data needed to evaluate the conclusions in the paper are present in the paper, can be accessed as tables and are provided with this paper. Experimental data and simulations are archived on the servers at LULI and are available from the corresponding author upon request. Source data are provided with this paper.

## Code availability

The code used to generate the simulations is an arbitrary kinetic algorithm (AKA). It is detailed in the Methods and available for download at https://zenodo.org/records/10435108.

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

## Acknowledgements

We acknowledge useful discussions with S. Orlando (INAF), T. Vinci (LULI), and A. Strugarek (CEA). We thank the LULI teams for their expert technical support. The authors also acknowledge support from the members of the VULCAN laser, engineering, and target fabrication group of the CLF, STFC, UK. This work was supported by the European Research Council (ERC) under the European Union's Horizon 2020 research and innovation program (Grant Agreement No. 787539; J.F.). The authors also acknowledge support from the Engineering and Physical Sciences Research Council (EP/P010059/1, M.B.) and the IMPULSE project by the European Union Framework Program for Research and Innovation Horizon 2020 (grant agreement no. 871161, M.B.). T.W. acknowledges the financial support of the IdEx University of Bordeaux/ Grand Research Program "GPR LIGHT". C. F. acknowledges support from AWE plc.

## Author contributions

J.F., A.Sl., M.S., and A.So. conceived the project. J.F., C.F., A.McI., S.N.C., H.A., P.M., T.W., and R.L. performed the experiments. J.F., W.Y., C.F., A.F.A.B., S.N.C., S.P., T.W., and E.D.F. analysed the data, with discussions with P.A., E.d'H., A.C., and M.B. The hybrid simulations were performed by A.Sl. The paper was mainly written by J.F., S.N.C., A.Sl., C.F., and W.Y. All authors commented and revised the manuscript.

## Competing interests

The authors declare no competing interests.
