## [Transparent Peer Review file · Nature Communications]

Saturation of the compression of two interacting magnetized plasma toroids evidenced in the laboratory

Corresponding Author: Dr Julien Fuchs

Version 0:

Reviewer comments:

Reviewer #1

(Remarks to the Author)

This study describes laboratory experiment and supporting numerical simulation of the magnetic field compression due to colliding of two flux tubes. Authors focus on the effect of reduction of field strength grow due to plasma flow breaking by the enhanced magnetic field. This is an interesting effect, that has been observed in-situ and documented for the Earth's magnetosphere (e.g., 10.1029/2018JA025588; I would suggest Authors to check this study for possible useful analogy). Authors also provide quite comprehensive review of space/astrophysical systems where such effect can be important. The numerical simulation reasonably confirms the main conclusions from the laboratory setup. I think this is high-level study, that can be considered for publication. However, there are a set of questions to be addressed before paper can be recommend for publication:

[1] Introduction should provide more detailed overview of the laboratory experiment: plasma beta, electron/ion temperature, sound and Alfvén Mach number, configuration of the magnetic field lines.

[2] page 3; what does this phrase mean? "This deflection is asymmetric because compression occurs only in a small region". Some explanation should be provided.

[3] page 4; role of $V_{\text{flow}} \times B$ field is not clear from the Authors' explanation.

[4] why formation of the current (magnetic) sheet would confirm some beta range?

[5] if inertial length and Larmor radius are comparable, then ion beta is about 1. But Authors use V_{flow} for evaluation of ion beta; why? Does this mean that thermal and flow speeds are comparable?

[6] what is the relation between symmetry of simulation and strength of compression?

Reviewer #2

(Remarks to the Author)

Please see uploaded PDF file

Reviewer #3

(Remarks to the Author)

Review of Nature Communications manuscript NCOMMS-24-18954-T:

As a space plasma physicist primarily working with the in-situ spacecraft measurements, I don't have expertise in experimental plasma physics. I am not particularly impressed by the significance of the reported work though. The compression of various magnetic field and plasma properties in space due to various processes or interactions is commonplace, and its mechanisms are well known, especially in the MHD regime and mostly in one spatial dimension. So perhaps the merits of a lab experiment are to probe deeper into the kinetic regime and in multiple dimensions. The current work hinted on those aspects but did not provide adequate details. Overall I am not inclined to recommend publication of this manuscript in Nature.

First of all, the applicability of the lab results for the space plasma processes has to be demonstrated, given the vast difference in the two systems.

- What are the noteworthy results?

>> The kinetic effects as observed and simulated, which needs to be expanded.

- Will the work be of significance to the field and related fields? How does it compare to the established literature? If the work is not original, please provide relevant references.

>> Generally No. The diagnostic measures as presented are probably of significance to the experimental plasma physics.

- Does the work support the conclusions and claims, or is additional evidence needed?

>> Generally yes.

- Are there any flaws in the data analysis, interpretation and conclusions? - Do these prohibit publication or require revision?

>> No.

- Is the methodology sound? Does the work meet the expected standards in your field?

>> The methods presented seem to be mostly related to diagnostic measures for probing or obtaining relevant parameters in a lab experiment. They have little significant novelty for the corresponding scientific investigations.

- Is there enough detail provided in the methods for the work to be reproduced?

>> Obviously it all depends on the reproducibility of the experimental setup by other facilities. Again I have no expertise to judge in that aspect.

Reviewer #4

(Remarks to the Author)

Review Report of NCOMMS 24- 18954-T:

“Saturation of the compression of two interacting magnetic flux tubes evidenced in the laboratory,”

by A. Sladkov, C. Fegan, W. Yao, A.F.A. Bott, S. N. Chen, H. Ahmed, E.D. Filippov,

R. Lelièvre, P. Martin, A. McIlvenny, T. Waltenspiel, P. Antici, M. Borghesi, S.

Pikuz, A. Ciardi, E. d’Humières, A. Soloviev, M. Starodubtsev, and J. Fuchs

In this paper, with the aid of laboratory experiments and hybrid numerical simulations mimicking the experiments, the authors examined a region where two plasmas interact with the magnetic field increased by compression. The authors reported that the compression of magnetic flux tubes is suppressed when the two opposite polarity magnetic fields encounter by plasma advection. The maximum compression rate defined by B_{\max}/B_{foot} is evaluated in the range of 3-5 for experiments, which is surprisingly small as compared to the values for the magnetic field compressed by a shock. It is argued that such a saturation of compression is caused by the magnetic pressure at the encounter point, and the electric field (electromotive force) due to the coupling of the plasma flow and magnetic field contributes to the suppression of compression through the re-direction of plasma flow.

I think this is a novel and interesting paper. This study directly showed the suppression of the magnetic-field compression for the first time by experiments. Their experiments are thoughtfully designed including the separation width between the two magnetic flux tubes, and performed in a well-organised manner. In this sense, the experimental results in this paper seem to be reliable. The physical interpretation of the suppression of compression due to the electric field (electromotive force) generated by the inflow and magnetic field is simple but appealing. However, as for the numerical setup and physical interpretation of the simulation results, I have a few comments and questions as follows.

1. Finite effective viscosity and diffusivity:

The authors wrote “the limited compression can be mostly understood in the frame of ideal MHD” (bottom of the right column of p. 3). In the numerics and theoretical arguments, the authors considered a system of ideal magnetohydrodynamics (MHD) since the Reynolds and magnetic Reynolds numbers are large as compared with unity. However, the ideal MHD is the limit of infinite Reynolds and magnetic Reynolds numbers. Then, it arises a natural question: How much Reynolds and magnetic Reynolds numbers (or their equivalents) do you have in the experiments?

2. Role of fluctuating velocity and magnetic fields:

If the Reynolds numbers are large in the sense that we can neglect the viscosity and magnetic diffusivity effects as compared to the non-linear terms, then turbulence and fluctuating fields are expected to show up due to non-linear coupling of modes. Then, the transport of the system is subject to the effective transports such as the turbulent viscosity and turbulent diffusivity. This is the case when we consider the large-scale fields. This means that, even if the small-scale MHD plasma magnetic fields show the property of well frozen-in to flows, the large-scale structures like the compression length may be subject to some turbulence or fluctuation effects. Is there any possibility for this experiment, some fluctuating fields play an essential role in compression suppression?

3. Density fluctuations:

Another point is related to the density fluctuation $\rho' (= \rho - \langle \rho \rangle)$ or density-variance ($\langle \rho'^2 \rangle$), where ρ is the density of plasma, $\langle \rho \rangle$ is appropriate average of density (mean density). In the presence of rapid compression with a strong mean-density gradient, it is expected that a large magnitude of density fluctuation represented by the density variance $\langle \rho'^2 \rangle$ is generated. Such a strong density variance may affect the transport of magnetic field through the turbulent electromotive force $\langle \mathbf{u}' \times \mathbf{b}' \rangle$. Is it possible to measure such quantity (density variance) in the experiments?

In the hybrid PIC simulation, the density variance should be represented by the variance of number density of ions $\langle n_{\text{ion}}^2 \rangle$.

4. Mach number dependence:

In the first paragraph of this paper, it was written as “two independent magnetic flux tubes (with field strength ~ 100 T [see Methods]), that counterpropagate toward each other at super-Alfvénic velocity” (in the left column of p. 1). In fig. 3(e), we see that the Mach number defined by the flow velocity normalised by the Alfvén velocity V_{flow}/V_A is in the range of $M_A < 0.5$. It is known that the effective transports, such as the momentum mixing, strongly depends on the Mach numbers [for example, the so-called Langley curve of high-speed flows, Samimy & Elliott, Phys. Fluids 28, 439 (1990); Papamoschou & Roshko, J. Fluid Mech. 197, 453 (1988); Goebel & Dutton, AIAA J. 29, 538 (1990)]. Is there any dependence of the compression suppression on the Mach or Alfvén Mach number of the flow velocity? Additional descriptions on this point are preferable from the viewpoint of identifying the physical mechanisms of compression suppression.

In summary, this is a novel and interesting work showing the real presence of the saturation or suppression of the compression of magnetic flux tubes in a quantitative manner. The experimental results themselves contain rich physical contents. In order to present more clear understanding of the physical mechanisms of this suppression, the presentations of the part on physical interpretations of results, in relation to the above points, should be improved by further descriptions by authors for the publication of the paper.

Version 1:

Reviewer comments:

Reviewer #1

(Remarks to the Author)

Authors have addressed all my questions and significantly improved description of the experiment setup. I'm ready to recommend this paper for publication

Reviewer #2

(Remarks to the Author)

I commend the authors for their careful attention to the reviewers' comments and recommendations regarding the original version of this manuscript. In light of the numerous revisions and clarifications in the revised version. I can now recommend publication in Nature Communications pending a few minor wording changes, as detailed below:

Upon rereading the Abstract (for which clarity is of paramount importance), I find a few sentences to be confusing and suggest the following rewording (or something similar) to avoid possible ambiguity—assuming it captures the intended meaning:

“Here we show, using laboratory experiments and matching three-dimensional hybrid simulations, that there is indeed a very effective saturation of the compression when two independent parallel-oriented magnetic fields regions encounter one another due to plasma advection. We found that the observed saturation is linked to a build-up of the magnetic pressure, which decelerates and redirects the inflows at their encounter point, thereby stopping further compression.”

There are also a few other minor wording issues:

In the caption to Fig. 1, the word “magnetic” appears twice in succession.

Near the bottom of the left column on p. 2, change “experimenta” to “experiments.” (Changing “ejecta” to “ejections” would also conform better to standard usage, but is not as serious a problem.)

In the right column of p. 3, “is no more azimuthally symmetric” would be better as “is no longer azimuthally symmetric.”

Near the bottom of the right column on p. 6, “This would allow to observe” should be “This would allow one to observe.”

As this list may not be exhaustive, a final careful proofreading is, as always, a good idea.

Reviewer #4

(Remarks to the Author)

Review Report of NCOMMS 24- 18954A:

“Saturation of the compression of two interacting magnetized plasma toroids evidenced in the laboratory,” by A. Sladkov, C. Fegan, W. Yao, A.F.A. Bott, S. N. Chen, H. Ahmed, E.D. Filippov, R. Lelièvre, P. Martin, A. McIlvenny, T. Waltenspiel, P. Antici, M. Borghesi, S. Pikuz, A. Ciardi, E. d'Humières, A. Soloviev, M. Starodubtsev, and J. Fuchs

In this revised manuscript, the authors carefully and elaborately addressed the issues had been pointed out by the

reviewers.

I think the authors successfully clarified and correct ambiguous expressions. Especially, explicit presentation of the physical parameters including the Reynolds, magnetic Reynolds, Peclet, sound Mach, Alfvén Mach numbers, etc. in the present experiment in comparison with the coronal mass ejection (CME) is very informative (Table 1).

I just have minor points:

As for my comments on the role of turbulence, velocity fluctuation, and density fluctuation, the authors replied as follows (on p.18, Reply to Reviewer #4):

"(I)n this specific experiment, we believe that the flows simply have not existed for long enough for turbulence to have developed by the time at which the compression of the field is observed. A very simple estimate of the characteristic time t_{turb} required for the development of turbulence can be made based on the in-flow velocity and the characteristic size of the compressed region: $t_{\text{turb}} \sim L/V = 6 \text{ ns}$. This time scale, which is essentially the timescale of nonlinear interactions between fluid motions in the compressed layer, is a factor of 5-6 times longer than the timescale on which peak compression is observed, and a factor of nearly 3 times longer than the time of the latest measurement. Furthermore, the good agreement between the experiment and the simulations, the latter of which do not manifest turbulent flows, supports the claim that turbulence does not seem to have a major impact on the compression."

"That being said, similarly to the (lack of) development of velocity fluctuations, we suspect that significant density fluctuations are unlikely to have had time to develop during the time scale over which the compression occurs in the experiment. Significant density fluctuations were also not observed in the simulations; the density was almost constant in the compression region, with a variance on the level of $\sim 0.01 \langle \rho \rangle$ (see the figure below)."

These points (too short timescale for turbulence evolution in the current case) should be explicitly written somewhere appropriate in the main text in a very brief manner.

This is because, in the beginning of Abstract, the authors referred to the anisotropisation of turbulent fields and resultant coherent structure formation in turbulence in the Universe as a crucial role of advected magnetic fields. This statement suggests to the reader that the transport due to turbulence will be argued in the present paper.

I also note the following points: The density variance of 0.01-0.1 means fluctuation level of 10-30% of the mean density. This is not necessarily small. At the same time, for instance, coronal mass ejection events are expected to be subjected to the turbulent process.

In summary, this is a novel and interesting work showing the real presence of the saturation or suppression of the compression of magnetised plasma toroids in a quantitative manner. The experimental results themselves contain rich physical contents. If the authors further addressed minor points (a brief description on the fluctuation effects) I mentioned above, the paper can be accepted for the publication.

I recommend the publication of this work.

Reviewer #1 (Remarks to the Author):

This study describes laboratory experiment and supporting numerical simulation of the magnetic field compression due to colliding of two flux tubes. Authors focus on the effect of reduction of field strength grow due to plasma flow breaking by the enhanced magnetic field. This is an interesting effect, that has been observed in-situ and documented for the Earth's magnetosphere (e.g., 10.1029/2018JA025588; I would suggest Authors to check this study for possible useful analogy). Authors also provide quite comprehensive review of space/astrophysical systems where such effect can be important. The numerical simulation reasonably confirms the main conclusions from the laboratory setup. I think this is high-level study, that can be considered for publication. However, there are a set of questions to be addressed before paper can be recommend for publication:

We thank the reviewer for the encouragement, we hope that all the issues discussed are now clarified satisfactorily. We also thank the reviewer for the reference, which indeed enriches the discussion and applicability of our study.

[1] Introduction should provide more detailed overview of the laboratory experiment: plasma beta, electron/ion temperature, sound and Alfvén Mach number, configuration of the magnetic field lines.

We agree with the reviewer that such an overview would improve the paper, and thus we now add two tables. A first one, in the main body of the paper, synthesizes the main parameters of both the laboratory plasma and of a relevant space plasma (both for the compressed magnetic sheet). The latter is detailed in Scolini et al. (2020). It pertains to satellite observations of a solar coronal mass ejection (CME) catching up with previously launched CMEs and inducing magnetic compression in a layer. A second table, in the (new) supplementary information, details the same plasma parameters, as well as the dimensional and dimensionless that can be derived, in order to fully characterize the plasmas.

The first Table is reproduced below (the equations defining all the parameters are described in detail in the revised paper, as is the expected magnetic compression ratio, based on the formula $B_{\max} = V_{\text{inflow}} \sqrt{(\mu_0 m_i n)}$ which is derived in the paper). Note that the parameter M_A is calculated with the magnetic field of the foot region, which is 60 T for the laboratory experiment and 10 nT for Scolini et al. (2020).

	Coronal mass ejection interaction	Present case
Inflow velocity (km/s)	250	50
B (T)	3e-8	450
Electron density (cm ⁻³)	2	1.3e21
Sound Mach number	1.4	3

Alfven Mach number	1.6	10
Characteristic spatial scale (cm)	1.5e12	3e-2
Reynolds number	5e5	1.2e3
Magnetic Reynolds number	1.5e17	63
Peclet number	1e4	3.5
Measured magnetic compression ratio	2.8	8
Expected magnetic compression ratio	3.2	7
Thermal beta	0.18	0.1
Euler number	1.8	3.7

We can see that in both cases, the magnetic Reynolds number (the ratio of the convection over ohmic dissipation) as well as the Peclet number (the ratio of magnetic convection to the magnetic diffusion) are larger than one. Thus, the two are in the same regime.

The fact that we can compare the two systems is based on (Ryutov 2018), by looking at the two scaling quantities, the Euler number ($Eu = V(\rho/p)^{1/2}$) and the thermal plasma beta ($\beta = 8\pi p/B^2$), where ρ is the mass density, $p = k_B(n_i n_{Ti} + n_e n_{Te})$). These two quantities have to be similar in order to have two systems scaled to each other and evolve in the same manner. We can see from the Table that indeed for both systems these numbers are quite close. From this, we can deduce that the two systems can indeed be compared to each other.

The Tables are now added to the paper, as well as a detailed discussion of the plasma parameters, see p.2 of the revised paper).

[2] page 3; what does this phrase mean? “This deflection is asymmetric because compression occurs only in a small region”. Some explanation should be provided.

The reviewer is right that the sentence was ambiguous, due to the use of the wording “asymmetric”. What we meant is to explain that, contrary to the proton fluence images that are azimuthally symmetric in the case of having a single plasma (whether on target 1 or 2), the proton image, in the case of having the two plasmas, is now azimuthally asymmetric, being expanded up-down, compared to left-right. We have thus modified the text as follows (on p.3):

“Compared to the single beam case shown in Fig.2 (a1) and (a2), the proton deflection pattern we observe in Fig.2 (a3) is no more azimuthally symmetric because compression occurs only in a small region”

We hope that the sentence is now clearer.

[3] page 4; role of $V_{\text{flow}} \times B$ field is not clear from the Authors' explanation.

We have now added more details into clarifying the origin and chain of events leading to the redirection of the plasma flows, triggered by the growth of the inductive electric field. This is now added on p.5:

“This field arises following the penetration of each plasma plume in the perpendicular (i.e. along the y – axis) magnetic field of the opposite plume. This then induces a $-V_{\text{flow}} \times B$ electric field which is directed along the s – axis (i.e. parallel and antiparallel to it), see Figure 4 (a) This field affects the plasma transport, redirecting the flows, in a similar manner as documented in space plasmas by satellite observation near Earth [Gabrielse 2019]. Then, the plasmas flowing along the s – axis also induce an electric field, illustrated by the red arrows in Figure 4 (b). The resulting plasma $E \times B$ drift is directed up/down along the slanted compressed magnetic sheet (i.e. along the s – axis). Since the induced electric field acts to deflect the inflows coming onto that sheet, it therefore limits further compression of the frozen-in magnetic field. ”

We hope this clarifies the physics behind limited magnetic compression and of the role of the inductive electric field.

[4] why formation of the current (magnetic) sheet would confirm some beta range?

We apologize for the unrigorous formulation and the non-adequate choice of “confirm”. What we wanted to describe in this paragraph is the magnetization of the electrons and ions when approaching the compressed magnetic sheet. Then, we wanted to quantify that magnetization, but our explanation was not well made, nor complete. We now have changed/detailed that paragraph as follows (on p.5):

“Figure 4 (c)-left panel shows that the electrons are highly magnetized near the magnetic sheet: there, the plasma beta parameter for the electrons (i.e. the ratio of the electron pressure to the magnetic field pressure) stays < 1 . This is not the case for the ions (see the right panel of Figure 4 (c)), as they transform their ram pressure into thermal pressure. This is shown in Figure 4 (d), which displays the ratio of the ion ram pressure ρV_{inflow}^2 , where ρ is the plasma density, to the magnetic pressure. The underlying mechanism is the deceleration of the ions by the electric field arising because of the gradient of the magnetic pressure. The ion magnetization is quantified in Figure 4 (e) (see the black dashed line). It shows that the ion Larmor radius is decreasing, and becomes, at the center of the compressed magnetic sheet, $R_L(r = 0) \approx d_i$, where d_i is the local ion inertial length.”

We hope this clarifies the issue.

[5] if inertial length and Larmor radius are comparable, then ion beta is about 1. But Authors use V_{flow} for evaluation of ion beta; why? Does this mean that thermal and flow speeds are comparable?

The reviewer is right that we use V_{inflow} to evaluate the ion beta, as it is the velocity which is responsible for the magnetic field transport i.e. it is the velocity that matters in the transport equation $dt(B) + dr(vr*B)=0$.

We can see straight away, by comparing the colors of the maps shown in Fig.4c (which displays the thermal beta of both the electrons and ions) and d (which displays the ion ram beta), that the ion flow is dominating over the ion thermal motion. Hence the use of V_{inflow} .

[6] what is the relation between symmetry of simulation and strength of compression?

To that question, we can state that the experiment is not perfectly symmetric, but that we nonetheless recover a similar compression ratio as the symmetric simulation. In the simulation, we can expect that the compression will be maximized, since the situation between the two incoming flows are perfectly symmetric.

In the experiment, we estimate the compression to be in the range 3-13 (i.e. with an average of 8) for the case shown in Fig.2 (and we now see, with the newly added figure showing a scan over time and over the parameter alpha, that such compression holds over many shots and many conditions) and in the simulation we have a compression ratio around 7.

So, the bottom line is that the observed magnetic compression is not that sensitive to the symmetry in the system, as is shown by the good agreement between the experiment (which cannot be perfectly symmetric) and the simulations (which is perfectly symmetric, and thus represents an ideal limit case).

We have now rephrased the relevant sections as follows (on p.6):

“To quantitatively evaluate B_{max} , we evaluate the point in the flow where the ions start to be magnetized, the flow velocity reduces and the magnetic field starts to grow. We evaluate such an edge of the compressed magnetic sheet to be located around $r=0.85d_0$, which corresponds to half of the sheet thickness. There, $V_{\text{inflow}} \approx 0.4$ (not shown). Since we also have $n_e/n_0 \sim 4$ and thus $n_i/n_0 = n_e/(Zn_0) \sim 0.2$, with $Z=18$ being the ionization of Cu (of $m_i = 64m_p$) at that location, we finally obtain $B_{\text{max}}/B_0 \sim 1.4$. Since $B_{\text{foot}}/B_0 = 0.2$, it then yields $B_{\text{max}}/B_{\text{foot}} \sim 7$, i.e. not only consistent with what is directly observed in the simulation (see Fig. 4.f), but also very close to the experimentally evaluated compression (~ 8).”

Reviewer #2 (Remarks to the Author):

This manuscript reports the results of a laboratory experiment in which two laser-irradiated magnetized plasma plumes expand into one another. The resulting compression of the magnetic field at the interface is studied to help understand the compression of magnetic fields in various space and astrophysical settings. In addition, a 3D hybrid simulation is used to model the experiment and aid in its interpretation. Laboratory experiment designed to model space and astrophysical plasma phenomena are important tools. This study could therefore be a worthy contribution to the field.

We thank the reviewer for the encouragement, we hope that all the issues discussed are now clarified satisfactorily.

There are nevertheless a number of issues in need of further clarification that I ask the authors to address before I could recommend publication in Nature Communications. These concerns are detailed in the following enumerated items:

1. In the title and throughout the manuscript the structures generate in the experiment are referred to as “flux tubes.” This terminology is problematical since the concept of a tube suggests a structure that is significantly larger in the axial than in the radial direction. The fields depicted in Fig. 1(b) do not fit this characterization. The experimental setup methods section refer to “plumes,” which would be a more apt descriptor (and one which I will use in this report). The first paragraph of the text makes reference to “. . . two independent magnetic flux tubes . . . , that counter-propagate toward each other. . . .” The term “propagate” suggest that the structures as a whole are moving toward one another. Would it not be more correct to say that they are expanding into one another ? I ask the authors to seriously reconsider their terminology. After all, the key feature is the magnetic field at the interface of the two expanding plumes—not their purported tube-like structure.

We agree with the reviewer. We used that term since it was advised to us by space physics colleagues, in relation to our magnetic structure being akin to a short section of a perfect “flux tube”. We now modify the text as follows:

The new title reads:

“Saturation of the compression of two interacting plasma-embedded magnetic fields evidenced in the laboratory”

We have otherwise replaced the “flux tube” everywhere, as e.g. in the caption of Fig.2, which now reads:

“Experimental proton-radiography images (see Methods) probing the single toroidal magnetic field in the plasma plume produced on (a1) target T2 or (a2) target T1.”

Just to give another example, one sentence on p.3 now reads:

“Just as in the experiments, we can simulate one single **plasma plume**, or the interaction between two **plasma plumes**.”

2. The experiment is set up so that at the interface between the expanding plumes the magnetic fields are strictly parallel to one another. In the terminology of magnetic reconnection, this would correspond to a shear angle of 0° , whereas anti-parallel reconnection would correspond to a shear angle of 180° (with intermediate shear angles corresponding to component or guide-field reconnection). Interacting regions in space and astrophysical settings presumably will occur over the full gamut of shear angles. While it is certainly legitimate to focus on a particular (albeit special) case of zero shear angle in the present study, the fact that this is a special case should be made clear—with a brief discussion of how relaxing this zero-shear condition (thereby allowing reconnection to occur) might affect the findings of this study.

We would like to point out that in fact we don't have a 0° shear because there is some curvature in each of the two plasma plumes. Moreover, since these are expanding (in all direction) plasma plumes, and since the situation is not perfectly symmetric, we cannot expect that in 3D the magnetic field lines, even at the contact point, would be in a perfect 0° shear configuration. Nevertheless, we would like to thank the reviewer for the suggestion for further insight in future experiments, where we could use elongated plasma plumes (such as done in <https://arxiv.org/pdf/2003.06351>), such that we can explore a more idealized 0° shear configuration, or exploit some target tilt (as we did in a previous paper investigating precisely reconnection in a guide field configuration, see <https://doi.org/10.1038/s41467-022-33813-9>) to add a guide field to the present compression configuration.

To clarify this issue, we now add the following sentence in the introduction (on p.1):

“Note that, due to the fact that the expanding plasmas do not have perfectly toroidal magnetic fields, and due to the fact that the two encountering plasmas are not perfectly symmetric, the encounter does not take place in a perfect 0 degree shear situation. Nonetheless, the fact, as detailed below, that we observe a similar compression of the magnetic fields when comparing the experimental results and the idealized simulations shows that the departure from the ideal 0° shear situation we aim at investigating here is not significant.”

and we also add the following to the conclusion (on p.6):

“Since the present investigation was set in a configuration where the two magnetic structures have field lines parallel to each other, i.e. with mostly zero shear angle, a direction to investigate other configurations encountered in space and astrophysical plasmas will be to observe the transition from magnetic compression, observed here, to reconnection [32], corresponding to a 180 degrees shear angle between the field lines. This would allow to observe the progressive weakening of the compression measured here to a growing annihilation of the magnetic field, being maximized [6] for perfectly

anti-parallel encountering magnetic fields. Doing this could be achieved simply by rotating the plane of one target with respect to the x-axis.”

3. In relation to the preceding item, the following statement appears on p. 4: “. . . the quasi-stationary compressed magnetic sheet can be very well approximated by the stationary solitary solution $\sim B_{\max} \cosh^{-2}(r/\lambda)$ (where λ is the width of the magnetic sheet) of the classical current sheet of the Harris equilibrium of magnetic reconnection [25].” The reference to a Harris sheet in this context is misleading given that the standard Harris sheet is a solution for an anti-parallel rather than a parallel magnetic configuration. More problematical is that $\cosh^{-2}(r/\lambda)$ is the functional form of the density profile in a Harris equilibrium, whereas the magnetic field is characterized by a $\tanh(r/\lambda)$ profile. It is reasonable to compare the measured magnetic profile with a $\cosh^{-2}(r/\lambda)$ approximation, but it is best to remove the reference to the Harris-sheet solution as a motivation.

We agree with the reviewer and have thus simplified the description of the fit as follows (on p.5):

“To estimate the compressed magnetic field strength B_{\max} , we first observe that, as shown by the full black and dotted red lines in Fig. 4 (e), the quasi-stationary compressed magnetic sheet can be very well approximated by the stationary solitary solution [21] $\sim 0.2 + B_{\text{fit}} \cosh^{-2}(r/\lambda)$, i.e. a compressed magnetic field peaked in the center (at $r=0$), on top of a foot at $0.2B_0$.”

4. The “raw optical interferometry images” in Fig. 1 (c) and (d) may be readily interpretable to those who regularly work with such diagnostics. However, I struggle to understand what is being depicted by the ‘zebra’ patterns—as, I suspect may be the case for many other readers. Please include additional guidance; or, preferably, recast these figures into a form that is easier to decipher.

We apologize for the lack of clarity. Indeed, the ‘zebra’ patterns are interference fringes imposed by the Mach-Zehnder setup we used to be able to quantitatively retrieve the plasma density in the areas that are not too dense. These allow to encode the line-integrated optical refraction index of the medium (the plasma) the probing laser beam passes through. A regular fringe pattern (equidistant fringes) means that the probe beam passes through vacuum (this is the configuration in which the interferometer is aligned). Perturbed fringes (with smaller or larger distances between fringes) mean that a medium with a non-zero refraction index is present on the path of the probe beam. From the quantitative measurement of these perturbations, we can come back to the quantitative measure of the density of the plasma the probe beam goes through.

The zones that are dark are due to the fact that when the plasma density gradient is too steep, the probing laser beam is refracted away, and hence no light can reach the imaging camera. We have clarified those issues by adding the following text to the caption of Fig.1:

“The dark zones correspond to the initial targets and target holders, as well as the dense parts of the expanding plasmas, following the laser interaction. The fringe pattern is related to the Mach-Zehnder setup [1] we used. The deformations of the fringes encode the line-integrated plasma density accumulated as the probing laser beam propagates along the z-axis. As the gradient of the plasma

increases toward the solid targets, the refraction of the probing laser beam increases until it cannot be collected anymore by the imaging optics, inducing the observed dark zones.”

5. There is reference on p. 3 to “shot-to-shot variation.” Please be more specific. The fact that shots #69 and #68 are plotted in Fig. 2(a5) suggest that the inventory of numbered runs is quite extensive. How many of these shots are for the same initial conditions as #69 and #68 (unlike, for example, #14 and #15)? Are #69 and #68 typical for this subset of runs, or are they in some sense best-case examples? Can you quantify the statistical variation for all such runs?

Indeed, we could obtain, despite the lasers we used being quite low-repetition rate, quite many data, showing the robustness of the observed magnetic field compression. To showcase this robustness and the observed statistical variations, we now add a new figure, Fig.3, to the paper.

And we add the following text to discuss the new Fig.3 (on p.4-5):

“For example, Fig. 3 shows variations of the raw ratio of the magnetic field strength measured at the center, in between the two interacting toroidal magnetic fields, over the incoming magnetic field (as measured in the outer part of the toroids). The variations correspond to probing the plasmas, and the interactions between the two toroids, at various times after the start of the laser irradiation of the targets (see Fig. 3 (a)) and to probing the interaction for various separation between the two targets (see Fig. 3 (b)). Note that the values of the raw ratio is between ~ 3 and ~ 9 , which translates into an average value for B_{\max}/B_{foot} ranging between ~ 7 and 25 .”

6. While Figs. 2(a4 & a5) for the experiment and (b4 & b5) for the simulation appear qualitatively similar, there are nevertheless significant difference. One notable difference is that in the experiment the in-plane magnetic field (integrated over z) becomes very small in the vicinity of the O-points, as expected. This is not the case for the simulation, where the minima are a significant fraction of the central maximum. Since the in-plane $|B|$ must vanish at an O-point, these elevated minima presumably are due to the fact that the O-points (in the x - y plane) are not aligned in the z direction. Is this, in fact, the case? If so, this is somewhat surprising since one might expect greater variability along z in the experiment than in the simulation. Please elaborate on these differences.

We thank the reviewer for noticing this! It stemmed from a double error on our part. First the label of the ordinates of Fig.2.a5 was wrong, in the sense that we wrote it as the integral of the absolute value of $B_{xy}dz$, when in fact what is shown is the absolute value of the integral of B_{xy} (which can thus have zero value). This is now corrected. The second error was to plot for the simulations (Fig.2.b5) indeed the integral of the absolute value of $B_{xy}dz$ (which does not go through zero), but which is not in correspondence to the experimental result. This is now corrected as well, with both experimental and simulation plots, showing the absolute value of the integral of $B_{xy}dz$. In this case, we indeed observed that, as expected, both plots nearing zero at the “O-points”, i.e. the center of the magnetic toroids, as can be seen in the new panels of Fig.2.

7. While the curves in Figs. 2(a5) and 2(b5) appear roughly similar in width and height, it is hard to make a quantitative comparison between them due to the difference in units (μm vs d_0 for length and kG cm vs $B_0 d_0$ for integrated field strength, respectively, for the experiment and the simulation). Using the values $B_0=400$ T and $8d_0 \sim 90 \mu\text{m}$ in the Methods section for the code on p. 8, the conversion would be $1 \text{ kG cm} \sim 0.22 B_0 d_0$.

I therefore highly recommend plotting the experimental values in scale units (d_0 and $B_0 d_0$) in panels (4a) and (5a) to allow for a direct comparison with the simulations. (The conversion to physical units can be added to the caption.) As the quantitative agreement between experiment and simulation will not be as strong as it currently appears, possible sources for the discrepancy should be discussed in the text.

Following the reviewer's suggestion, we have added double scales for all the plots of Fig.2 pertaining to the simulation results. This is based on the following conversion, as detailed in the Methods:

$$B_0 = 300 \text{ [T]}$$

$$n_0 = 1.3e+21 \text{ [cm}^{-3}\text{]}$$

$$T_0 = 171 \text{ [eV]}$$

$$2ns \rightarrow t\Omega_i = 17$$

$$\text{Ion inertial length } d_0 = 11.6 \text{ [\mu m]}$$

The conversion of $d_0 b_0$ to real quantities has a factor 3.5.

As detailed in the text, we have to make a compromise in order for the simulation to be sustainable computationally, and this was done by reducing by a factor ~ 2 the size of the magnetic toroids and the distance between the two toroids. Thus, we also have some quantitative difference e.g. in the absolute value of the compressed B by a factor ~ 2 , but as detailed in the discussion, the compression ratio itself, i.e. $B_{\text{max}}/B_{\text{foot}}$ is well matched between the experiment and the simulation.

8. It is not completely clear from the text where the location of “the foot of the magnetic sheet” is being measured. Please include—on one or more of the figures—a marker indicating the assumed position of the foot.

The “foot” basically designates the magnetic field outside of the compression layer. It is now clearly indicated in Fig. 4.e by arrows. It is where the B field has its asymptotic value of 0.2, following our approximation $0.2 + \cosh^{-1} \dots$, i.e. where the B field is unperturbed.

9. Since a primary goal of this study is to explain why the compression of the magnetic field is limited, it would be helpful to specify the compression ratio that would be expected for the experimental configuration in the absence of the suppression mechanism proposed.

The plot of the plasma electron β_e in Fig. 3(c) shows a large decrease at the interface between the two expanding plasma plumes. (Why not plot $\beta_e + \beta_i$ rather than β_e alone?) This minimum implies that the magnetic field (or, more correctly, $P_B \sim B^2$) experiences a much larger compression factor than the embedding plasma itself. Even for the larger ram pressure of Fig. 3(d), there is presumably a value for

the magnetic compression such that the magnetic pressure at the the interface balances the total plasma pressure (thermal plus ram). What is this compression ratio? In this regard, it would be helpful to supplement Fig. 3 with a superimposed set of line plots (along x at $y = 0$) showing the magnetic pressure, the plasma thermal pressure, and the plasma ram pressure.

To answer the reviewer, we present below a plot of various quantities: the electron ($P_{\perp e}$ - black dashed) and ion ($P_{\perp i}$ - black solid) thermal pressures, the magnetic pressure (black dotted) and the ion ram pressure (red solid). Overlaid are also the ion (blue) and electron (green) densities. These quantities are all plotted along the axis r , which is (see Fig.4.a of the paper), the axis perpendicular to the compressed magnetic sheet.

We can see that the ram pressure drops at the mid-plane of the compressed magnetic sheet, which can be expected since the inflow plasmas are redirected to being parallel to the compressed magnetic sheet. The electron pressure drops to zero because it is expelled by the magnetic pressure. However, the ion pressure stays high because of the compression effect: as plasma decelerates approaching the mid-plane, its temperature grows. The magnetic field accumulation is a consequence of the finite ion flow velocity in the magnetic field transport equation, while the chaotic temperature motion does not contribute to the magnetic field growth, which is why we do not see a perfect pressure equilibrium.

Additionally, we thank the reviewer for suggesting to estimate the magnetic compression from a pressure balance, or equivalently from energy conservation law. We have now added such estimate in the paper, as follows (on p.6):

“Alternatively, we can derive the value of the compressed magnetic field by considering that the energy is conserved in the system. Initially, the energy is partitioned between that of the unperturbed magnetic field with pressure $0.5B_{\text{foot}}^2$ and that of the plasma with ram pressure ρV_{in}^2 . In the compressed magnetic sheet, all the energy is mostly contained within the magnetic field, with pressure $0.5(B_{\text{max}})^2$. Thus, the energy conservation yields $B_{\text{max}}/B_{\text{foot}} = \sqrt{1 + 2M_A^2}$ where $M_A = V_{\text{in}}/V_A$ and $V_A = B_{\text{foot}}/\sqrt{(m_i n_i)}$. Since we have $MA \sim 10$ for the inflow (see Fig. 4.e), we can deduce a maximum magnetic compression ratio of $B_{\text{max}}/B_{\text{foot}} \sim 14$, i.e. of the same order as the previous estimate.”

10. In addition to the technical concerns enumerated above, the following stylistic issues should also be addressed:

(a) Since the Abstract should be a self-standing synopsis of the study, the inclusion of 14 citations is at odds with this goal. Surveying a random sample of abstracts of papers published in Nature Communications turned up no examples that included citations. Whether or not such citations are consistent with the journal’s editorial policy, I strongly encourage the authors to nevertheless move them to the body of the text—even if that requires a small additional amount of repetition.

The reviewer is right, the citations in the abstract have now been removed.

(b) While this manuscript largely avoids overstatement, reference to “the Universe at all possible scales” (Abstract) and “all scales in the Universe” (bottom of p. 4) comes off as hyperbolic. Elsewhere, the applicability of this research is given a more limited scope—for example, in the following text from p. 5: “Indeed, when two plasmas carrying magnetic field interact, as e.g. in configurations as varied as galaxy-cluster [3], supernova remnant-cloud [39] or wind-exoplanet [40] configurations. . .” Changing “all possible scales” to “a diverse range of scales” (or something similar) would still convey the intended meaning without overstating the case.

Indeed, this has now changed, following your suggestion.

11. Finally, please proofread for minor typographical errors such as the following: (a) On p. 1 (right column) “. . . magnetic field distribution in strength and spatial in the xy-plane” should presumably read “. . . spatial distribution. . .” (b) In the caption to Fig. 1 (line 4) “x-axis” should read “z-axis.” (c) The caption to Fig. (4) contains no reference to panel (c)

Thank you for spotting these, they are now corrected.

Despite the need for substantial clarification, the experimental analysis appears to be fundamentally sound and should be relevant to the study of the interaction of magnetized plasma volumes in space and astrophysical contexts. The simulations are also appropriate, although the differences between the experimental and simulation results need further reconciliation. One potential strength of the side-by-side comparison of experiment and simulation is that, once the two are fully reconciled, further exploration of parameter space should be easier to accomplish through modification of the simulation parameters than through modification of the experimental setup.

In summary, I encourage the authors to undertake the requested clarifications and revisions with the goal of eventual publication in Nature Communications.

We thank the reviewer for the encouragement, we hope that all the issues discussed are now clarified satisfactorily.

Reviewer #3 (Remarks to the Author):

As a space plasma physicist primarily working with the in-situ spacecraft measurements, I don't have expertise in experimental plasma physics. I am not particularly impressed by the significance of the reported work though. The compression of various magnetic field and plasma properties in space due to various processes or interactions is commonplace, and its mechanisms are well known, especially in the MHD regime and mostly in one spatial dimension. So perhaps the merits of a lab experiment are to probe deeper into the kinetic regime and in multiple dimensions. The current work hinted on those aspects but did not provide adequate details. Overall I am not inclined to recommend publication of this manuscript in Nature.

First of all, the applicability of the lab results for the space plasma processes has to be demonstrated, given the vast difference in the two systems.

The reviewer is right that the laboratory plasma and space plasmas are vastly different. However, one can still look at analogy, taking advantage of a possible rescaling between the two. Extensive work has been performed regarding this question, in particular in a set of papers by D. Ryutov (most notably Ryutov, D. D., Drake, R. P., & Remington, B. A. 2000, *The Astrophysical Journal Supplement Series*, 127, 465 and Ryutov, D. D. 2018, *Physics of Plasmas*, 25, 100501). Following the reviewer suggestion, and that of reviewer 1, we have now added a Table in the paper. It synthesizes the main parameters of both the laboratory plasma and of a relevant space plasma, i.e. that detailed in Scolini et al. (2020), of a solar coronal mass ejecta (CME) catching up with previously launched CMEs and inducing magnetic compression in a layer.

Based on (Ryutov 2018), we compare the two scaling quantities, the Euler number ($Eu = V(\rho/p)^{1/2}$) and the thermal plasma beta ($\beta = 8\pi\rho/B^2$), where ρ is the mass density, $p = k_B(n_iT_i + n_eT_e)$). These two quantities have to be similar in order to have two systems scaled to each other and evolve in the same manner. We can see from the Table 1 in the paper that indeed for both systems these numbers are quite close. From this, we can deduce that the two systems can indeed be compared to each other.

- What are the noteworthy results?

>> The kinetic effects as observed and simulated, which needs to be expanded.

We have now indeed expanded the discussion pertaining to the underlying physics. Following this, we hope that the referee finds the discussion more satisfying.

We now add the following discussion of the results (the modified text is also in red in the paper, see p.5 of the paper):

“As we will now detail, the limited compression can be mostly understood in the frame of ideal MHD. Fundamentally, that limitation is the consequence of an induced electric field ($V_{inflow} \times B$, in which V_{inflow} is the flow velocity and B is the magnetic field) that is present on both sides of the compressed magnetic sheet. This field arises following the penetration of each plasma plume in the magnetic field of the

opposite plume. This then induces a $V_{\text{flow}} \times B$ electric field which is directed up/down, along the s-axis shown in Figure 4 (a). This field affects the plasma transport, redirecting the flows, in a similar manner as documented in space plasmas by satellite observation near Earth [20]. Then, the plasmas flowing along the s-axis also induce an electric field, illustrated by the red arrows in Figure 4 (b). The resulting plasma drift $E \times B$ is directed up/down along the slanted compressed magnetic sheet (i.e. along the s-axis). Since the induced electric field acts to deflect the inflows coming onto that sheet, it therefore limits further compression of the frozen-in magnetic field."

- Will the work be of significance to the field and related fields? How does it compare to the established literature? If the work is not original, please provide relevant references.

>> Generally No. The diagnostic measures as presented are probably of significance to the experimental plasma physics.

We would first like to point out to the new Table 1, where we detail the comparison between the laboratory plasma and the one of a collision between two CMEs, as detailed in Scolini et al. (see also the answers to point 1 of reviewer #1). We can see from the Table that the two systems can indeed be compared to each other.

Beyond this, we would like to point out that we believe our work to be indeed relevant outside of the experimental plasma physics that we use to highlight the physics at play during magnetic field compression.

Indeed, it is known that in general magnetic fields play a crucial role in many processes, such as the formation of stars [Krumholz2019/10.3389/fspas.2019.00007, Bracco2020/10.1051/0004-6361/202039282] and galaxies [Müller2020/10.1038/s41550-020-01234-7], the dynamics of accretion disks [Zamponi2021/10.1093/mnras/stab2657] or solar phenomena.

In many configurations, magnetic field compression occurs, caused e.g. in the clouds by supernova shockwaves, stellar feedback from massive stars, gravitational collapse, or cloud-cloud collisions. Understanding how the magnetic fields are compressed in these regions can be essential for elucidating the mechanisms driving star formation. Moreover, systems like active galactic nuclei, X-ray binaries, or young stellar objects are characterized by magnetized accretion disks [Seifried2011/10.1111/j.1365-2966.2011.19320.x]. The compression of magnetic fields within these disks might be caused, for instance, by differential rotation, shearing motion, magneto-rotational instability. In this case the field compression affects the dynamics of accretion and, consequently, the multi-wavelength emission observed. The compression of magnetic fields can play an important role also in the Sun's atmosphere, particularly in regions such as sunspots and solar flares, and might influence solar activity and space weather phenomena [Owens2013/10.12942/lrsp-2013-5]. In this case, studying magnetic field compression in solar physics can be crucial for understanding solar dynamics and their impact on Earth's environment [Belakhovsky2017/10.1186/s40623-017-0696-1]. Finally, magnetic fields pervade the interstellar medium (ISM), affecting the dynamics of interstellar gas clouds and cosmic ray propagation [Strong2007/10.1146/annurev.nucl.57.090506.123011]. Compression of magnetic fields in regions of ISM turbulence or shock fronts (as, for instance, in supernova remnants) can also have significant implications for the evolution of galaxies and the interstellar environment.

In short, we indeed use laboratory plasmas to highlight the dynamics at play in the compression between two magnetic field structures, but we believe that what we reveal applies to a wide range of physical problems in the Universe where such effects will also impact the dynamics of the systems. It has been already amply demonstrated that, by the approach of “laboratory astrophysics”, a quantitative understanding of fundamental astrophysical phenomena can be attained through laboratory experiments [Takabe2021/10.1017/hpl.2021.35]. Our investigation uses this general approach, applied here to investigate the particular problem of magnetic field compression in plasmas.

- Does the work support the conclusions and claims, or is additional evidence needed?

>> Generally yes.

Thank you for the positive appreciation of our work

- Are there any flaws in the data analysis, interpretation and conclusions? - Do these prohibit publication or require revision?

>> No.

Thank you for the positive appreciation of our work

- Is the methodology sound? Does the work meet the expected standards in your field?

>> The methods presented seem to be mostly related to diagnostic measures for probing or obtaining relevant parameters in a lab experiment. They have little significant novelty for the corresponding scientific investigations.

We would like to point out that indeed use very sturdy and well-assessed diagnostics that are commonly used in order to retrieve a maximum of observables. These are:

*optical interferometry to record the plasma density (see Fig.1.c-d)

*proton radiography to record the magnetic field distributions (see Fig.2)

And these are complemented by state-of-the-art hybrid simulations to retrieve the plasma parameters that are not directly accessible to the experiment (like the flow velocity).

- Is there enough detail provided in the methods for the work to be reproduced?

>> Obviously it all depends on the reproducibility of the experimental setup by other facilities. Again I have no expertise to judge in that aspect.

In answer to your query, and to the point 5 raised by reviewer 2, we have added a new Fig.3, showing the robustness of the observed magnetic field compression. This new Fig.3 shows the increase of the lin-integrated magnetic field, as recorded in the raw data, for many shots, scanning the time as well as the separation between the targets. It showcases the robustness and the observed statistical variations of our experimental measurements.

Reviewer #4 (Remarks to the Author):

In this paper, with the aid of laboratory experiments and hybrid numerical simulations mimicking the experiments, the authors examined a region where two plasmas interact with the magnetic field increased by compression. The authors reported that the compression of magnetic flux tubes is suppressed when the two opposite polarity magnetic fields encounter by plasma advection. The maximum compression rate defined by B_{\max}/B_{foot} is evaluated in the range of 3-5 for experiments, which is surprisingly small as compared to the values for the magnetic field compressed by a shock. It is argued that such a saturation of compression is caused by the magnetic pressure at the encounter point, and the electric field (electromotive force) due to the coupling of the plasma flow and magnetic field contributes to the suppression of compression through the re-direction of plasma flow.

I think this is a novel and interesting paper. This study directly showed the suppression of the magnetic-field compression for the first time by experiments. Their experiments are thoughtfully designed including the separation width between the two magnetic flux tubes, and performed in a well-organised manner. In this sense, the experimental results in this paper seem to be reliable. The physical interpretation of the suppression of compression due to the electric field (electromotive force) generated by the inflow and magnetic field is simple but appealing.

We thank the reviewer for the encouragement, we hope that all the issues discussed are now clarified satisfactorily.

However, as for the numerical setup and physical interpretation of the simulation results, I have a few comments and questions as follows.

1. Finite effective viscosity and diffusivity:

The authors wrote “the limited compression can be mostly understood in the frame of ideal MHD” (bottom of the right column of p. 3). In the numerics and theoretical arguments, the authors considered a system of ideal magnetohydrodynamics (MHD) since the Reynolds and magnetic Reynolds numbers are large as compared with unity. However, the ideal MHD is the limit of infinite Reynolds and magnetic Reynolds numbers. Then, it arises a natural question: How much Reynolds and magnetic Reynolds numbers (or their equivalents) do you have in the experiments?

Following the reviewer’s suggestion, and that of reviewers 1 and 3, we have now added a Table in the paper. It synthesizes the main parameters of both the laboratory plasma and of a relevant space plasma, i.e. that detailed in Scolini et al. (2020), of a solar coronal mass ejecta (CME) catching up with previously launched CMEs and inducing magnetic compression in a layer.

The Table gives the Reynolds ($1.2e3$) and magnetic Reynolds numbers (63) for the laboratory plasma, which indeed justifies the ideal MHD framework for the investigated laboratory plasma.

2. Role of fluctuating velocity and magnetic fields:

If the Reynolds numbers are large in the sense that we can neglect the viscosity and magnetic diffusivity effects as compared to the non-linear terms, then turbulence and fluctuating fields are expected to show up due to non-linear coupling of modes. Then, the transport of the system is subject to the effective transports such as the turbulent viscosity and turbulent diffusivity. This is the case when we consider the large-scale fields. This means that, even if the small-scale MHD plasma magnetic fields show the property of well frozen-in to flows, the large-scale structures like the compression length may be subject to some turbulence or fluctuation effects. Is there any possibility for this experiment, some fluctuating fields play an essential role in compression suppression?

The reviewer is correct to say that, in general, turbulent transport could play a role in affecting the degree of compression of magnetic fields in colliding flows. However, in this specific experiment, we believe that the flows simply have not existed for long enough for turbulence to have developed by the time at which the compression of the field is observed. A very simple estimate of the characteristic time t_{turb} required for the development of turbulence can be made based on the in-flow velocity and the characteristic size of the compressed region: $t_{\text{turb}} \sim L/V = 6 \text{ ns}$. This time scale, which is essentially the timescale of nonlinear interactions between fluid motions in the compressed layer, is a factor of 5-6 times longer than the timescale on which peak compression is observed, and a factor of nearly 3 times longer than the time of the latest measurement. Furthermore, the good agreement between the experiment and the simulations, the latter of which do not manifest turbulent flows, supports the claim that turbulence does not seem to have a major impact on the compression.

3. Density fluctuations:

Another point is related to the density fluctuation ρ' ($= \rho - \langle \rho \rangle$) or density-variance ($\langle \rho'^2 \rangle$), where ρ is the density of plasma, $\langle \rho \rangle$ is appropriate average of density (mean density). In the presence of rapid compression with a strong mean-density gradient, it is expected that a large magnitude of density fluctuation represented by the density variance $\langle \rho'^2 \rangle$ is generated. Such a strong density variance may affect the transport of magnetic field through the turbulent electromotive force $\langle \mathbf{u}' \times \mathbf{b}' \rangle$. Is it possible to measure such quantity (density variance) in the experiments? In the hybrid PIC simulation, the density variance should be represented by the variance of number density of ions $\langle n_{\text{ion}}^2 \rangle$.

In the experiment, the interferometry did not show direct evidence of density fluctuations, although we cannot rule out fluctuations on smaller scales than the resolution of this diagnostic directly. That being said, similarly to the (lack of) development of velocity fluctuations, we suspect that significant density fluctuations are unlikely to have had time to develop during the time scale over which the compression occurs in the experiment. Significant density fluctuations were also not observed in the simulations; the density was almost constant in the compression region, with a variance on the level of $\sim 0.01 \langle \rho \rangle$ (see the figure below).

4. Mach number dependence:

In the first paragraph of this paper, it was written as “two independent magnetic flux tubes (with field strength ~ 100 T [see Methods]), that counterpropagate toward each other at super-Alfvénic velocity” (in the left column of p. 1). In fig. 3(e), we see that the Mach number defined by the flow velocity normalised by the Alfvén velocity V_{flow}/V_A is in the range of $M_A < 0.5$. It is known that the effective transports, such as the momentum mixing, strongly depends on the Mach numbers [for example, the so-called Langley curve of high-speed flows, Samimy & Elliott, Phys. Fluids 28, 439 (1990); Papamoschou & Roshko, J. Fluid Mech. 197, 453 (1988); Goebel & Dutton, AIAA J. 29, 538 (1990)]. Is there any dependence of the compression suppression on the Mach or Alfvén Mach number of the flow velocity? Additional descriptions on this point are preferable from the viewpoint of identifying the physical mechanisms of compression suppression.

We thank the reviewer for the occasion to clarify our results. In fact, what was shown was the velocity, but in units of V_A calculated on the base of the normalization B_0 and n_0 factors, which do not necessarily represent the *local* V_A (since B is changing over space). To make things clear, we do not show the velocity anymore, but directly MA (i.e. taking into account the local B in the calculation) in Fig.4.e. This now clearly shows that the inflows have $MA \sim 10$ when approaching the interaction region.

Regarding the second point raised by the reviewer, we thank him/her for the suggestion. We have now added a discussion on the dependence of the compression on the Alfvén Mach number. It is as follows (on p.6):

“Alternatively, we can derive the value of the compressed magnetic field by considering that the energy is conserved in the system. Initially, the energy is partitioned between that of the unperturbed magnetic field with pressure $0.5B_{\text{foot}}^2$ and that of the plasma with ram pressure ρV^2 inflow. In the compressed magnetic sheet, all the energy is mostly contained within the magnetic field, with pressure $0.5(B_{\text{max}})^2$. Thus, the energy conservation yields $B_{\text{max}}/B_{\text{foot}} = \sqrt{1 + 2M_A^2}$ where $MA = V_{\text{inflow}}/V_A$ and $V_A =$

$B_{\text{foot}}/\sqrt{\mu_0} \text{mini}$. Since we have $MA \sim 10$ for the inflow (see Fig. 4.e), we can deduce a maximum magnetic compression ratio of $B_{\text{max}}/B_{\text{foot}} \sim 14$, i.e. of the same order as the previous estimate.”

In summary, this is a novel and interesting work showing the real presence of the saturation or suppression of the compression of magnetic flux tubes in a quantitative manner. The experimental results themselves contain rich physical contents. In order to present more clear understanding of the physical mechanisms of this suppression, the presentations of the part on physical interpretations of results, in relation to the above points, should be improved by further descriptions by authors for the publication of the paper.

We thank the reviewer for the encouragement, we hope that all the issues discussed are now clarified satisfactorily.

REVIEWERS' COMMENTS

Reviewer #1 (Remarks to the Author):

Authors have addressed all my questions and significantly improved description of the experiment setup. I'm ready to recommend this paper for publication

Thank you

Reviewer #2 (Remarks to the Author):

I commend the authors for their careful attention to the reviewers' comments and recommendations regarding the original version of this manuscript. In light of the numerous revisions and clarifications in the revised version. I can now recommend publication in Nature Communications pending a few minor wording changes, as detailed below:

Upon rereading the Abstract (for which clarity is of paramount importance), I find a few sentences to be confusing and suggest the following rewording (or something similar) to avoid possible ambiguity—assuming it captures the intended meaning:

“Here we show, using laboratory experiments and matching three-dimensional hybrid simulations, that there is indeed a very effective saturation of the compression when two independent parallel-oriented magnetic fields regions encounter one another due to plasma advection. We found that the observed saturation is linked to a build-up of the magnetic pressure, which decelerates and redirects the inflows at their encounter point, thereby stopping further compression.”

This has been changed, as suggested by the reviewer, see the text in red in the paper.

There are also a few other minor wording issues:

In the caption to Fig. 1, the word “magnetic” appears twice in succession.

Near the bottom of the left column on p. 2, change “experimenta” to “experiments.” (Changing “ejecta” to “ejections” would also conform better to standard usage, but is not as serious a problem.)

In the right column of p. 3, “is no more azimuthally symmetric” would be better as “is no longer azimuthally symmetric.”

Near the bottom of the right column on p. 6, “This would allow to observe” should be “This would allow one to observe.”

All this has been changed, as suggested by the reviewer, see the text in red in the paper.

As this list may not be exhaustive, a final careful proofreading is, as always, a good idea.

Reviewer #2 (Remarks on code availability):

The code provides an adequate README file and should be implementable by the interested reader with sufficient familiarity with C++ and Python. However, I have not attempted to build a working version myself.

Reviewer #4 (Remarks to the Author):

Review Report of NCOMMS 24- 18954A:

“Saturation of the compression of two interacting magnetized plasma toroids evidenced in the laboratory,”

by A. Sladkov, C. Fegan, W. Yao, A.F.A. Bott, S. N. Chen, H. Ahmed, E.D. Filippov, R. Leli`evre, P. Martin, A. McIlvenny, T. Waltenspiel, P. Antici, M. Borghesi, S. Pikuz, A. Ciardi, E. d’Humi`eres, A. Soloviev, M. Starodubtsev, and J. Fuchs

In this revised manuscript, the authors carefully and elaborately addressed the issues had been pointed out by the reviewers.

I think the authors successfully clarified and correct ambiguous expressions. Especially, explicit presentation of the physical parameters including the Reynolds, magnetic Reynolds, Peclet, sound Mach, Alfvén Mach numbers, etc. in the present experiment in comparison with the coronal mass ejection (CME) is very informative (Table 1).

I just have minor points:

As for my comments on the role of turbulence, velocity fluctuation, and density fluctuation, the authors replied as follows (on p.18, Reply to Reviewer #4):

"(I)n this specific experiment, we believe that the flows simply have not existed for long enough for turbulence to have developed by the time at which the compression of the field is observed. A very simple estimate of the characteristic time t_{turb} required for the development of turbulence can be made based on the in-flow velocity and the characteristic size of the compressed region: $t_{\text{turb}} \sim L/V = 6 \text{ ns}$. This time scale, which is essentially the timescale of nonlinear interactions between fluid motions in the compressed layer, is a factor of 5-6 times longer than the timescale on which peak compression is observed, and a factor of nearly 3 times longer than the time of the latest measurement. Furthermore, the good agreement between the experiment and the simulations, the latter of which do not manifest turbulent flows, supports the claim that turbulence does not seem to have a major impact on the compression."

"That being said, similarly to the (lack of) development of velocity fluctuations, we suspect that significant density fluctuations are unlikely to have had time to develop during the time scale over which the compression occurs in the experiment. Significant density fluctuations were also not observed in the simulations; the density was almost constant in the compression region, with

a variance on the level of ~ 0.01 $\langle \rho \rangle$ (see the figure below)."

These points (too short timescale for turbulence evolution in the current case) should be explicitly written somewhere appropriate in the main text in a very brief manner.

This is because, in the beginning of Abstract, the authors referred to the anisotropisation of turbulent fields and resultant coherent structure formation in turbulence in the Universe as a crucial role of advected magnetic fields. This statement suggests to the reader that the transport due to turbulence will be argued in the present paper.

I also note the following points: The density variance of 0.01-0.1 means fluctuation level of 10-30% of the mean density. This is not necessarily small. At the same time, for instance, coronal mass ejection events are expected to be subjected to the turbulent process.

The text relative to the lack of turbulence in the plasmas of the study has now been added to the paper, as suggested by the reviewer, see the text in red in the paper.

In summary, this is a novel and interesting work showing the real presence of the saturation or suppression of the compression of magnetised plasma toroids in a quantitative manner. The experimental results themselves contain rich physical contents. If the authors further addressed minor points (a brief description on the fluctuation effects) I mentioned above, the paper can be accepted for the publication.

I recommend the publication of this work.

Reviewer #4 (Remarks on code availability):

I could not get the web page with the URL:
<https://zenodo.org/records/10435108> (2023)
"Page not found"

We have checked and indeed the page can be accessed, and the code found.

Review of *Nature Communications* Manuscript NCOMMS-24-18954-T

“Saturation of the compression of two interacting magnetic flux tubes
evidenced in the laboratory”

A. Sladkov, C. Fegan, W. Yao, [15 other authors], and J. Fuchs

This manuscript reports the results of a laboratory experiment in which two laser-irradiated magnetized plasma plumes expand into one another. The resulting compression of the magnetic field at the interface is studied to help understand the compression of magnetic fields in various space and astrophysical settings. In addition, a 3D hybrid simulation is used to model the experiment and aid in its interpretation.

Laboratory experiment designed to model space and astrophysical plasma phenomena are important tools. This study could therefore be a worthy contribution to the field. There are nevertheless a number of issues in need of further clarification that I ask the authors to address before I could recommend publication in *Nature Communications*. These concerns are detailed in the following enumerated items:

1. In the title and throughout the manuscript the structures generate in the experiment are referred to as “flux tubes.” This terminology is problematical since the concept of a tube suggests a structure that is significantly larger in the axial than in the radial direction. The fields depicted in Fig. 1(b) do not fit this characterization. The experimental setup methods section refer to “plumes,” which would be a more apt descriptor (and one which I will use in this report). The first paragraph of the text makes reference to “. . . two independent magnetic flux tubes . . . , that counter-propagate toward each other. . . .” The term “propagate” suggest that the structures *as a whole* are moving toward one another. Would it not be more correct to say that they are *expanding into one another*? I ask the authors to seriously reconsider their terminology. After all, the key feature is the magnetic field at the interface of the two expanding plumes—not their purported tube-like structure.
2. The experiment is set up so that at the interface between the expanding plumes the magnetic fields are strictly parallel to one another. In the terminology of magnetic reconnection, this would correspond to a shear angle of 0° , whereas *anti-parallel* reconnection would correspond to a shear angle of 180° (with intermediate shear angles corresponding to *component* or *guide-field* reconnection). Interacting regions in space and astrophysical settings presumably will occur over the full gamut of shear angles. While it is certainly legitimate to focus on a particular (albeit special) case of zero shear angle in the present study, the fact that this *is* a special case should be made clear—with a brief discussion of how relaxing this zero-shear condition (thereby allowing reconnection to occur) might affect the findings of this study.
3. In relation to the preceding item, the following statement appears on p. 4: “. . . the quasi-stationary compressed magnetic sheet can be very well approximated by the stationary solitary solution $\sim B_{\max} \cosh^{-2}(r/\lambda)$ (where λ is the width of the magnetic sheet) of the classical current sheet of the Harris equilibrium of magnetic reconnection [25].” The

reference to a Harris sheet in this context is misleading given that the standard Harris sheet is a solution for an *anti-parallel* rather than a *parallel* magnetic configuration. More problematical is that $\cosh^{-2}(r/\lambda)$ is the functional form of the *density* profile in a Harris equilibrium, whereas the magnetic field is characterized by a $\tanh(r/\lambda)$ profile. It is reasonable to compare the measured magnetic profile with a $\cosh^{-2}(r/\lambda)$ approximation, but it is best to remove the reference to the Harris-sheet solution as a motivation.

4. The “raw optical interferometry images” in Fig. 1 (c) and (d) may be readily interpretable to those who regularly work with such diagnostics. However, I struggle to understand what is being depicted by the ‘zebra’ patterns—as, I suspect may be the case for many other readers. Please include additional guidance; or, preferably, recast these figures into a form that is easier to decipher.
5. There is reference on p. 3 to “shot-to-shot variation.” Please be more specific. The fact that shots #69 and #68 are plotted in Fig. 2(a5) suggest that the inventory of numbered runs is quite extensive. How many of these shots are for the same initial conditions as #69 and #68 (unlike, for example, #14 and #15)? Are #69 and #68 typical for this subset of runs, or are they in some sense best-case examples? Can you quantify the statistical variation for all such runs?
6. While Figs. 2(a4 & a5) for the experiment and (b4 & b5) for the simulation appear qualitatively similar, there are nevertheless significant difference. One notable difference is that in the experiment the in-plane magnetic field (integrated over z) becomes very small in the vicinity of the O-points, as expected. This is not the case for the simulation, where the minima are a significant fraction of the central maximum. Since the in-plane $|B|$ must vanish at an O-point, these elevated minima presumably are due to the fact that the O-points (in the x - y plane) are not aligned in the z direction. Is this, in fact, the case? If so, this is somewhat surprising since one might expect greater variability along z in the experiment than in the simulation. Please elaborate on these differences.
7. While the curves in Figs. 2(a5) and 2(b5) appear roughly similar in width and height, it is hard to make a *quantitative* comparison between them due to the difference in units (μm vs d_0 for length and kG cm vs B_0d_0 for integrated field strength, respectively, for the experiment and the simulation). Using the values $B_0=400$ T and $8d_0 \sim 90 \mu\text{m}$ in the Methods section for the code on p. 8, the conversion would be $1 \text{ kG cm} \sim 0.22 B_0d_0$. I therefore highly recommend plotting the experimental values in scale units (d_0 and B_0d_0) in panels (4a) and (5a) to allow for a direct comparison with the simulations. (The conversion to physical units can be added to the caption.) As the *quantitative* agreement between experiment and simulation will not be as strong as it currently appears, possible sources for the discrepancy should be discussed in the text.
8. It is not completely clear from the text where the location of “the foot of the magnetic sheet” is being measured. Please include—on one or more of the figures—a marker indicating the assumed position of the foot.
9. Since a primary goal of this study is to explain why the compression of the magnetic field is limited, it would be helpful to specify the compression ratio that *would be expected* for the experimental configuration in the absence of the suppression mechanism proposed.

The plot of the plasma electron β_e in Fig. 3(c) shows a large decrease at the interface between the two expanding plasma plumes. (Why not plot $\beta_e + \beta_i$ rather than β_e alone?) This minimum implies that the magnetic field (or, more correctly, $P_B \sim B^2$) experiences a much larger compression factor than the embedding plasma itself. Even for the larger ram pressure of Fig. 3(d), there is presumably a value for the magnetic compression such that the magnetic pressure at the the interface balances the *total* plasma pressure (thermal plus ram). What is this compression ratio? In this regard, it would be helpful to supplement Fig. 3 with a superimposed set of line plots (along x at $y = 0$) showing the magnetic pressure, the plasma thermal pressure, and the plasma ram pressure.

10. In addition to the technical concerns enumerated above, the following stylistic issues should also be addressed:
 - (a) Since the Abstract should be a self-standing synopsis of the study, the inclusion of 14 citations is at odds with this goal. Surveying a random sample of abstracts of papers published in *Nature Communications* turned up no examples that included citations. Whether or not such citations are consistent with the journal’s editorial policy, I strongly encourage the authors to nevertheless move them to the body of the text—even if that requires a small additional amount of repetition.
 - (b) While this manuscript largely avoids overstatement, reference to “the Universe at all possible scales” (Abstract) and “all scales in the Universe” (bottom of p. 4) comes off as hyperbolic. Elsewhere, the applicability of this research is given a more limited scope—for example, in the following text from p. 5: “Indeed, when two plasmas carrying magnetic field interact, as e.g. in configurations as varied as galaxy-cluster [3], supernova remnant-cloud [39] or wind-exoplanet [40] configurations. . .” Changing “all possible scales” to “a diverse range of scales” (or something similar) would still convey the intended meaning without overstating the case.
11. Finally, please proofread for minor typographical errors such as the following:
 - (a) On p. 1 (right column) “. . . magnetic field distribution in strength and spatial in the xy -plane” should presumably read “. . . spatial distribution. . .”
 - (b) In the caption to Fig. 1 (line 4) “ x -axis” should read “ z -axis.”
 - (c) The caption to Fig. (4) contains no reference to panel (c)

Despite the need for substantial clarification, the experimental analysis appears to be fundamentally sound and should be relevant to the study of the interaction of magnetized plasma volumes in space and astrophysical contexts. The simulations are also appropriate, although the differences between the experimental and simulation results need further reconciliation. One potential strength of the side-by-side comparison of experiment and simulation is that, once the two are fully reconciled, further exploration of parameter space should be easier to accomplish through modification of the simulation parameters than through modification of the experimental setup.

In summary, I encourage the authors to undertake the requested clarifications and revisions with the goal of eventual publication in *Nature Communications*.